# MixACM: Mixup-Based Robustness Transfer via Distillation of Activated Channel Maps

**Muhammad Awais** [1, 2]*†, **Fengwei Zhou** [1]*, **Chuanlong Xie** [1]*, **Jiawei Li** [1],
**Sung-Ho Bae** [2]‡, **Zhenguo Li** [1]

[1] Huawei Noah's Ark Lab
[2] Department of Computer Science, Kyung-Hee University, South Korea
awais@khu.ac.kr, {zhoufengwei, xie.chuanlong, li.jiawei}@huawei.com,
shbae@khu.ac.kr, li.zhenguo@huawei.com

## Abstract

Deep neural networks are susceptible to adversarially crafted, small and impercep­tible changes in the natural inputs. The most effective defense mechanism against these examples is adversarial training which constructs adversarial examples during training by iterative maximization of loss. The model is then trained to minimize the loss on these constructed examples. This min-max optimization requires more data, larger capacity models, and additional computing resources. It also degrades the standard generalization performance of a model. Can we achieve robustness more efficiently? In this work, we explore this question from the perspective of knowledge transfer. First, we theoretically show the transferability of robustness from an adversarially trained teacher model to a student model with the help of mixup augmentation. Second, we propose a novel robustness transfer method called Mixup-Based Activated Channel Maps (MixACM) Transfer. MixACM transfers robustness from a robust teacher to a student by matching activated channel maps generated without expensive adversarial perturbations. Finally, extensive experi­ments on multiple datasets and different learning scenarios show our method can transfer robustness while also improving generalization on natural images.

## 1  Introduction

Deep learning models have achieved impressive performance on a wide variety of challenging tasks such as image recognition, natural language generation, game playing, etc. However, these models are susceptible to even small changes in the input space. It is possible to craft tiny changes in the input such that a model classifies unaltered input correctly but classifies the same input incorrectly after small and visually imperceptible perturbations [71, 27]. These altered inputs are known as adversarial examples, and this vulnerability of the state-of-the-art models has raised serious concerns [41, 58, 10, 47].

Many defense mechanisms have been proposed to train deep models to be robust against such adversarial perturbations. These techniques include defensive distillation [57], gradient regularization [29, 58, 62], model compression [21, 45], activation pruning [23, 59], adversarial training [49], etc. Among them, Adversarial Training (AT) is one general strategy that is the most effective [4].

---

*Equal Contribution.

†This work was carried out at Huawei Noah's Ark Lab. The webpage for the project is available at: awais-rauf.github.io/MixACM

‡Corresponding Author.

35th Conference on Neural Information Processing Systems (NeurIPS 2021).

In general, adversarial training is a kind of data augmentation technique that trains a model on examples augmented with adversarial changes [49]. The adversarial training consists of an inner, iterative maximization loop to augment natural examples with adversarial perturbations, and an outer minimization loop similar to normal training. Many different methods have been introduced to improve robustness [76, 15, 52, 91, 79, 66, 94, 78, 7, 81, 39, 44, 56], but all of them are fundamentally based on the principle of training on adversarially augmented examples.

Adversarial training is more challenging compared with normal training. Better robust generalization requires larger capacity models [49, 53, 80]. Even over-parameterized models that can easily fit data for normal training [92] may have insufficient capacity to fit adversarially augmented data [97]. The sample complexity of adversarial training can be significantly higher than normal training, and it requires more labeled [64] or unlabeled data [76, 15, 52, 91]. The inner maximization for the generation of adversarial perturbations for adversarial training is significantly more expensive computationally as it requires iterative gradient steps with respect to the inputs (e.g., adversarial training takes $\sim$7x more time compared with normal training). Adversarial training also degrades the performance of a model on natural examples significantly [53, 94].

Can we attain robustness more efficiently, i.e., with less data, small capacity models, without extra back-propagation steps, and at no significant sacrifice of clean accuracy? For normal training, knowledge transfer is one possible paradigm for the efficient training of deep neural networks. In normal knowledge transfer, a pre-trained teacher model is leveraged to efficiently train a student model on the same or similar dataset [70]. However, knowledge transfer methods are designed to transfer features related to normal generalization which may be at odds with robustness [74, 38]. A recent line of work has explored transferring or distilling robustness from pre-trained models [32, 67, 25, 16]. Although these techniques are effective, they also require extra gradient computation and may not work in some cases.

In this work, we argue that a better approach for robustness transfer is to distill intermediate features of a robust teacher, generated on mixup examples. Our proposed approach does not require any additional gradient computation and works well with smaller models and fewer data samples. It can also effectively transfer robustness across models and datasets.

We begin with the theoretical analysis and show that adversarial loss of a student model can be bounded with two terms: natural loss and distance between the student and teacher model on mixup examples. To minimize the distance between robust teacher and student, we proposed a new distillation method. Our proposed distillation method is based on channel-wise activation analysis of robustness by Bai et al. [7] and other studies showing that adversarially trained models learn fundamentally different features [38, 74, 99].

Channels of a convolutional neural network learn different disentangled representations which, when combined, describe specific semantic concepts [8]. Samples of different classes activate different channels for an intermediate layer. However, adversarial examples make these channels activate more uniformly thereby destroying class-related information. Adversarial training solves this issue by forcing a similar channel-activation pattern for normal and adversarial examples [7]. Our proposed distillation method generates activated channel maps from a robust teacher which has already learned robust channel-activation patterns. The student, then, is forced to match these activated channel maps.

We have conducted extensive experiments to show the effectiveness of our method using various datasets and under different learning settings. We start by exhibiting the ability of our method to transfer robustness without requiring adversarial examples. We then show that our method can distill robustness from large pre-trained models to smaller models and it can transfer robustness across datasets. We also show that our method is capable to robustify a model even with a small number of examples. Concisely, our contributions are as follows:

1. We show that adversarial loss of a student model can be bounded with the distance between robust teacher and student model on mixup examples.

2. We propose a new method to transfer robustness from the intermediate features of robust pre-trained models.

3. We have demonstrated effectiveness of our method with experiments on CIFAR-10, CIFAR-100, and ImageNet datasets. We showed that MixACM can achieve adversarial robustness comparable to the state-of-the-art methods without generating adversarial examples. Our method also improves clean accuracy significantly.

## 2 Related Work

**Adversarial Training and Robust Features.** Adversarial attacks are considered to be a security concern [9, 71, 27, 14], and many methods have been proposed to defend against such attacks [29, 27, 57, 30, 58, 73, 49, 12, 43, 62, 21, 45, 23, 59]. Adversarial training [49] is the most effective defense [4] among these methods. The fundamental principle of adversarial training is to generate adversarial perturbation during training by doing iterative back-propagation w.r.t. input. The model is then trained on these perturbed examples. Building upon this principle, different aspects of adversarial training have extensively been studied, e.g., effect of large unlabeled data [76, 15, 91], theoretical aspects to improve the trade-off between accuracy and robustness [94], improving parts of adversarial training for better robustness [78, 81, 56, 7, 28], making adversarial training fast [52, 66, 79, 1], pruning with adversarial training [87, 65], adversarial training with unlabeled data [39], etc. Our work is different from these methods as our main concern is robustness transfer without generating adversarial perturbations. Another line of work studied the effects of adversarial training and showed that adversarial training learns 'fundamentally different' [53] features compared with normal training [38, 86, 100]. Our method is motivated by these studies and is designed to leverage these robust features.

**Knowledge and Robustness Transfer.** Knowledge transfer is training a student model more efficiently on a dataset with the help of a teacher model trained on a similar or related dataset [70]. Many different settings have been explored to achieve this objective by using soft labels produced by teacher [11, 35], knowledge learned by intermediate layers of teacher also called feature distillation [61, 89, 37, 88, 75, 40, 68, 34, 33, 72], or matching input gradients [89, 70]. However, the main objective of these knowledge transfer methods is improving the generalization on natural, unperturbed images. Our work is fundamentally different from these works as we want to transfer robustness.

Our work is motivated by [32, 67, 25, 16, 6] showing transferability of robustness. Hendrycks et al. [32] showed that robust features learned on large datasets like ImageNet can improve adversarial training. Shafahi et al. [66] showed that robust pre-trained models can act as feature extractors in transfer learning. These two methods are limited in application as pre-trained models are used as backbone models. In [25], the authors showed that the robustness can be distilled more efficiently from a large pre-trained teacher model to a smaller student model by using the teacher-produced class scores on adversarial examples. Compared with [25], our method does not require any adversarial training. In [16], the distillation is performed by matching the gradient of the teacher and student, requiring fine-tuning teacher on target dataset and back-propagation to get gradients of the teacher and student w.r.t. inputs and training of a discriminator. Compared with [16], our proposed method does not require any back-propagation steps in addition to those carried out in normal training.

**Activation and Robustness.** Many works have explored the role of activations in the robustness from different angles, such as modifying output of intermediate layers to improve adversarial training [82, 23, 48, 51, 84], using different activation functions [59, 77, 85, 28, 55, 7, 5] or interpretation of what is learned by adversarial training with activations [86, 7]. However, we leveraged activations to transfer robustness. Our distillation method is inspired by the channel-wise analysis of robustness proposed by Bai et al. [7].

**Mixup Augmentation and Robustness.** Mixup augmentation was introduced to improve the generalization however, it also improved performance against one-step adversarial attacks [95]. This motivated many mixup-based adversarial defense methods, e.g., Pang et al. [54], Lee et al. [42], Archambault et al. [3], Bunk et al. [13], Chen et al. [17]. These defense methods primarily improve adversarial training by mixup. Our work, however, is different as we explore the role of the mixup in robustness transfer from both theoretical and algorithmic perspectives. Our theoretical work is based on [98] that showed a connection between mixup and robustness theoretically. Goldblum et al. [25] also briefly empirically explored the effect of some augmentation methods on robustness transfer.

## 3 Setup

We consider task of mapping input $x \in \mathcal{X} \subseteq \mathbb{R}^d$ to label $y \in \mathcal{Y} = \{1, 2, \ldots, K\}$. Given the training data $\mathcal{D} = \{(x_1, y_1), ..., (x_n, y_n)\}$, the goal is to learn a classifier $f : \mathcal{X} \to \mathbb{R}^k$ from a hypothesis space $\mathcal{F}$. Let $L(f(x), y)$ be the loss function that measures that how poorly the classifier $f(x)$

predicts the label $y$. Suppose $f$ can be decomposed into $g \circ h$, where $h \in \mathcal{H}$ is a feature extractor and $g \in \mathcal{G}$ stands for the classifier on top of feature extractor.

**Adversarial Augmentation**: Adversarial training augments natural examples with adversarial perturbations which are attained by maximizing the loss, i.e., $\tilde{x} = x + \delta$ where $\delta = \arg\max_{\|\delta'\|_p \leq \epsilon} L\big(f(x + \delta'), y\big)$. Here $\|\cdot\|_p$ stands for $L_p$ norm. Throughout this paper, we take $p = \infty$. We denote the adversarial loss [49] as:

$$\tilde{\mathcal{L}}(f, \mathcal{D}) = \frac{1}{n} \sum_{i=1}^{n} \max_{\|\delta\|_\infty \leq \epsilon} L(f(x_i + \delta), y_i),$$

where $\epsilon$ is the perturbation budget that governs the adversarial robustness of the model.

**Mixup Augmentation**: Mixup constructs virtual examples by linearly combining two examples [54]: $\tilde{x}_{ij}(\lambda) = \lambda x_i + (1 - \lambda)x_j$, and $\tilde{y}_{ij}(\lambda) = \lambda y_i + (1 - \lambda)y_j$, where $\lambda \in [0, 1]$ follows the distribution $P_\lambda$. The mixup loss is

$$\mathcal{L}_{mix}(f, \mathcal{D}) = \frac{1}{n^2} \sum_{i=1}^{n} \sum_{j=1}^{n} \mathbb{E}_{\lambda \sim P_\lambda}[L(f(\tilde{x}_{ij}(\lambda)), \tilde{y}_{ij}(\lambda))].$$

# 4   Analysis: How to Transfer Robustness?

In this section, we provide an analysis on how to transfer robustness with knowledge distillation and mixup augmentation. We first explore this problem from a theoretical perspective and show that student's adversarial loss can be decomposed into the sum of the teacher's adversarial loss and the distillation loss. Furthermore, these two loss functions can be approximated by mixup-based augmentation. We then discuss how we can design a distillation loss that distills robustness from the intermediate features of a robust model.

## 4.1   Theoretical Analysis

**Robustness Transfer.** We give a generalization bound for the robust test error via distillation. Different to the classical bound, we compress the hypothetical space $\mathcal{F}$ into a smaller space derived by the teacher feature extractor. Hsu et al. [36] points out that the distillation helps to derive a fine-grained analysis for the prediction risk and avoid getting a vacuous generalization bound. Therefore, we take the following theorem as a starting point to derive robustness transfer. We consider the loss function $L(f(x), y) = 1 - \phi_\gamma\big(f(x), y\big)$, where $\phi_\gamma$ is the softmax function with temperature $\gamma > 0$:

$$\phi_\gamma\big(f(x), y\big) = \frac{\exp(f(x)_y/\gamma)}{\sum_{k=1}^{K} \exp(f(x)_k/\gamma)}.$$

**Theorem 4.1.** *Let the temperature $\gamma > 0$ be given. Then, with probability $1 - \delta$, for any $f \in \mathcal{F}$,*

$$\mathbb{E}\big[\max_{\|\delta\|_\infty \leq \epsilon} \mathbb{I}\{\hat{y}(x + \delta) \neq y\}\big] \leq 2\tilde{\mathcal{L}}(f, \mathcal{D}) + \frac{8}{\gamma}\tilde{\mathcal{L}}_{dis}(f, \mathcal{D}) + 4\mathfrak{R}_S(\Psi_T) + \frac{16}{n} + 6\sqrt{\frac{2/\delta}{2n}}, \quad (1)$$

*where $\mathfrak{R}_S$ is the Rademacher complexity, $\Psi_T = \{\max_{\|\delta\|_\infty < \epsilon} L(g \circ h^T(x + \delta), y), \ g \in \mathcal{G}\}$ and*

$$\tilde{\mathcal{L}}_{dis}(f, \mathcal{D}) = \frac{1}{n} \inf_{g \in \mathcal{G}} \sum_{i=1}^{n} \max_{\|\delta\|_\infty < \epsilon} \big|f(x_i + \delta) - g \circ h^T(x_i + \delta)\big|.$$

On the right hand side of (1), two terms depend on the classifier $f$, i.e., the adversarial loss $\tilde{\mathcal{L}}(f, \mathcal{D})$ and the distillation loss $\tilde{\mathcal{L}}_{dis}(f, \mathcal{D})$. Thus, we can minimize $2\tilde{\mathcal{L}}(f, \mathcal{D}) + 8\tilde{\mathcal{L}}_{dis}(f, \mathcal{D})/\gamma$ to find a classifier with small robust test error. Note that the loss function $L$ is bound above by the cross-entropy loss. So $\tilde{\mathcal{L}}(f, \mathcal{D})$ can be replaced with a common-used adversarial loss based on cross-entropy.

**Adversarial Robustness, Transfer and Mixup.** However, solving the maximization problem for $\delta$ is time-consuming, which is a severe problem in adversarial training especially for large models. For binary classification task with logistic loss function, Zhang et al. [98] proved that $\tilde{\mathcal{L}}(f, \mathcal{D})$ can be

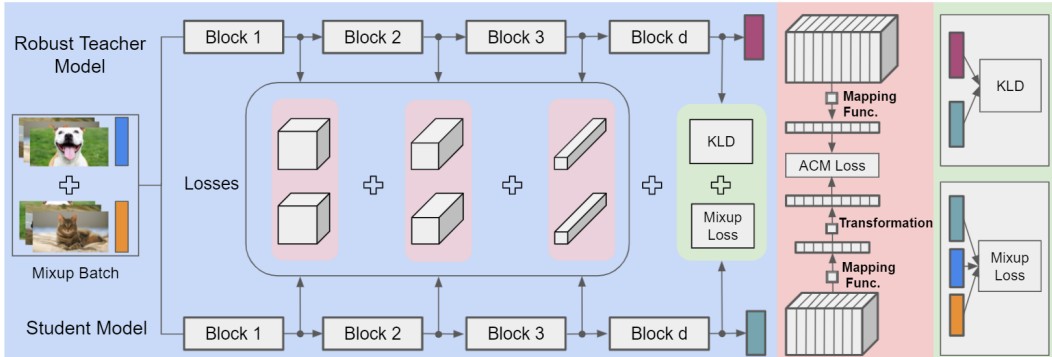

Figure 1: **Overview of our approach**. The mixup augmented examples are passed through a robust teacher and a student to get intermediate features. These features are then passed through a mapping function to get activated channel maps. The student mimics the activated channel maps of robust teacher by minimizing ACM loss. The student model also minimizes the standard KLD and Mixup loss.

bounded above by $\mathcal{L}_{mix}(f, \mathcal{D})$. In addition, the distillation loss that measures the distance between the student model and the teacher model also considers the worst-case $\delta$. In our algorithm, we use the KL divergence or Euclidean distance and mixup sample to design the distillation loss. Inspired by the Theorem 3.3 in [98], we present the following example to illustrate the rationality behind our design.

**Theorem 4.2.** *Suppose that $\mathcal{Y} = \{0, 1\}$ and $L$ is the logistic loss. Let $f$ be a fully connected neural network with ReLU activation function or max-pooling. Suppose a learnt teacher model $f^T$ with the same network structure is given. Denote $\mathcal{D}_{dis} = \{(x_i, f^T(x_i)), i = 1, \ldots, n\}$. Under certain assumptions, we have for any positive scalar $\alpha$,*

$$\tilde{\mathcal{L}}(f, \mathcal{D}) + \alpha \tilde{\mathcal{L}}(f, \mathcal{D}_{dis}) \leq \mathcal{L}_{mix}(f, \mathcal{D}) + \alpha \mathcal{L}_{mix}(f, \mathcal{D}_{dis}).$$

This result shows that both the adversarial loss and the distillation loss can be bounded above by their mixup loss. Please see Appendix.B for more details and complete proof. Here the distillation loss is formulated by the prediction accuracy for pseudo-labels generated by a learnt teacher model, which is also considered by [15].

## 4.2 Considerations for Algorithmic Design

In order to transfer robustness, our theoretical analysis suggests minimizing the distance between teacher and student on mixup examples. However, as observed by Goldblum et al. [25] and our experiments in Section 6, minimization of KL-Divergence between the outputs of teacher and student models is not enough. How can we reduce the gap between the teacher and student more effectively? Based on previous work [24, 86, 7], we hypothesize that the gap between latent space of robust teacher and student is large and it requires distillation from intermediate features.

To reduce the distance between the latent spaces of robust teacher and student, we get inspiration from recent channel-based analysis of robustness [7] and works on the interpretation of what is learned by robust models [53, 38, 86]. Specifically, we consider convolution channels of CNNs to be fundamental learning blocks: each channel learns class-specific patterns and the output of the model is a combination of these basic patterns. As noted by [38], these basic blocks learn both robust and non-robust features with natural training. Adversarial examples amplify non-robust features while suppressing the robust features, e.g., adversarial examples frequently activate channels that are rarely activated by natural examples [7]. Adversarial training works by suppressing non-robust channels and making the activation frequency of natural and adversarial examples similar.

Motivated by this analysis, we propose Activated Channel Map (ACM) transfer loss. Our method extracts the activated channel maps of the robust model's intermediate features on mixup examples. The student is then forced to mimic these maps. Specifically, we consider the maximum value of each channel as a proxy for the extent of how activated a channel is: a channel is less activated if it has a smaller maximum value and more activated if it has a high maximum value. We then make the student match the normalized activation map of the teacher. And, further analysis of our method is presented in the supplementary material. We present the core of our method in the next section.

# 5 Methodology

## 5.1 Activated Channels Maps (ACM) Transfer Loss

In this section, we introduce our Mixup-Based Robustness Transfer method. As discussed above, the method aims to transfer robustness from a robust teacher model $f^T$ to a student model $f^S$ via matching activated channel maps of a robust teacher on mixup augmented examples. An overview of our method is given in Figure 1.

The purpose of robust teacher is to supervise the student's focus on robust features. This is achieved by matching activated channel maps of teacher and student. The activated channel maps of a model is extracted as follows: given $i$-th block activations $\mathcal{A}_i$ (output of $i$-th layer), we apply a function $g_c : \mathbb{R}^{B \times C \times H \times W} \rightarrow \mathbb{R}^{B \times C \times 1 \times 1}$ to get the activated channel map $a_i = g_c(\mathcal{A}_i)$, as illustrated in Figure 2.

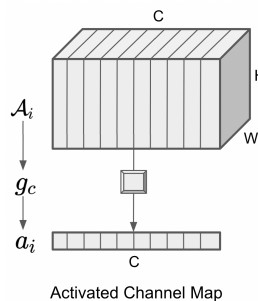

The activated channel map of student and teacher may differ in scale. The contribution of each channel in the final output, on the other hand, depends only on the relative value compared with other channels. Therefore, we first normalize this map. To force the student to mimic the teacher's activated channel maps, we use $\ell_2$ distance between the normalized activated channel maps of teacher and student. For $i$-th layer, we can define ACM loss as:

Figure 2: Mapping function $g_c$ is applied channelwise to get the Activated Channel Maps (ACM).

$$\mathcal{L}_{acm_i}(\mathcal{A}_i^T, \mathcal{A}_i^S) = \|\frac{a_i^T}{\|a_i^T\|_2} - \frac{a_i^S}{\|a_i^S\|_2}\|_2^2.$$

We minimize this distance at the end of each block of the models. A block is a set of convolution layers grouped like residual blocks in ResNets [31]. For a student and teacher model with $d$ such blocks, the total ACM loss becomes as follows: $\mathcal{L}_{acm}(x; f^T, f^S) = \sum_{i=0}^{d} \mathcal{L}_{acm_i}(f^T(x)_i, f^S(x)_i)$, where $f(x)_i$ represents output of the model at $i$-th block.

**Mapping Function** ($g_c$): The aim of activated channel map function ($g_c$) is to extract a map of least to most activated channels. For this purpose, we used maximum value of each channel: $g_c(x) = \max_i x_i \forall i \in \{0, ..., c_l\}$, where $c_l$ is number of channels in $l$-th layer.

## 5.2 Transferring Robustness with Different Feature Sizes

Our method extracts a scalar value corresponding to each channel in a layer. Until now, we assume that the teacher and student have the same number of channels. For this case, we only need the $g_c$ mapping function. To accomplish knowledge transfer across models with a different number of channels, we need to apply transformation either on a teacher or student's activated channel maps. We experimented with two transformations: adaptive pooling and affine transformation.

**Adaptive Pooling**: Pooling is used in most CNN architectures to reduce the size of feature vectors and gain invariance to small transformations of the input. The pooling layer works by dividing the input into patches and taking an average or maximum value over these patches. By changing the size of these patches, pooling can also be used for up-sampling. Pooling operation can be described as follows: $y[p] = \max_{q \in \Omega(p)} x[q]$, where $\Omega$ represents a patch or neighbourhood of $x[p]$. We use PyTorch implementation of 1d adaptive maximum pooling to either down or up-sample teacher or student output.

**Affine Transformation**: We can also use the learning function to convert the input size to output size. For this, we use a fully connected layer. Based on our ablation studies described in supplementary material, we use adaptive pooling as a default in our method.

## 5.3 KL Divergence Loss with Soft Labels

Soft labels have shown to be useful for adversarial robustness by Pang et al. [55]. Therefore, for distillation on the same dataset (e.g., where teacher and student have the same classification layer), we minimize the KL divergence between the logits provided by teacher and student following [35, 25]. The KLD loss becomes as follows: $\mathcal{L}_{kld}(x; f^T, f^S) = \gamma^2 KL(y^T(\gamma), y^S(\gamma))$, where $\gamma$ is

Table 1: **Detailed Comparison**. CIFAR-10 test accuracy and robustness comparison under two different $\ell_\infty$ attacks of magnitude $\epsilon = 8/255$. Our method outperforms robustness distillation methods in both accuracy and robustness. It outperforms existing state-of-the-art robust models in clean accuracy while retaining comparable robustness. All reported values for our method are mean $\pm$ std of five repetitions. B.P. in the table stands for back propagation.

| Method | Model | Extra B.P. Steps | Clean Acc. | PGD 100 | Auto Attack |
|---|---|---|---|---|---|
| AT (Madry et al. [49]) | WRN-34-10 | ✓ | 87.14 | 47.04 | 44.04 |
| LAT (Singh et al. [69]) | WRN-34-10 | ✓ | 87.80 | 53.04 | 49.12 |
| TLA (Mao et al. [50]) | WRN-34-10 | ✓ | 86.21 | 50.03 | 47.41 |
| YOPO (Zhang et al. [93]) | WRN-34-10 | ✓ | 87.20 | 47.98 | 44.83 |
| Free AT (Shafahi et al. [66]) | WRN-34-10 | ✓ | 86.11 | 46.19 | 41.47 |
| TRADES (Zhang et al. [94]) | WRN-34-10 | ✓ | 84.92 | 56.43 | 53.08 |
| MART (Wang et al. [78] ) | WRN-34-10 | ✓ | 83.07 | 55.57 | - |
| AWP (Wu et al. [81]) | WRN-34-10 | ✓ | 85.36 | 59.12 | 56.17 |
| FAT (Zhang et al. [96]) | WRN-34-10 | ✓ | 84.52 | 54.36 | 53.51 |
| Overfitting (Rice et al. [60]) | WRN-34-20 | ✓ | 85.34 | 58.00 | 53.42 |
| BoT TRADES (Pang et al. [55]) | WRN-34-20 | ✓ | 86.43 | 54.39 | 54.39 |
| Adv. Pretraining (Hendrycks et al. [32]) | WRN-28-10 | ✓ | 87.10 | 57.40 | 54.92 |
| IGAM (Chan et al. [16] ) | WRN-34-10 | ✓ | 88.70 | 43.00 | - |
| RKD (Goldblum et al. [26]) | WRN-34-10 | ✗ | 89.38 | 0.21 | 0 |
| Ours (w/o Mixup) | WRN-34-10 | ✗ | 91.17±0.06 | 49.53±1.44 | 47.38±1.51 |
| Ours (w/ Mixup) | WRN-34-10 | ✗ | 90.76±0.05 | 56.65±0.93 | 53.93±0.77 |

the temperature parameter, and $y(\gamma)$ is output logits of a model normalized by temperature parameter as used by Hinton et al. [35]. We omit this loss term when transferring from one dataset to another.

## 5.4 Mixup Fueled Distillation

Our theoretical analysis suggest minimizing distance between teacher and student with mixup examples. To this end, we use mixup to linearly combine inputs and outputs and minimize loss on these inputs. The final training objective for the student is as follows:

$$\min_{f^S} \mathbb{E}_{(x,y)\sim\mathcal{D}} \mathbb{E}_{\lambda\sim P_\lambda} \left[ (1 - \alpha_{kld})\mathcal{L}_{mix} + \alpha_{kld}\mathcal{L}_{kld} + \alpha_{acm}\mathcal{L}_{acm} \right],$$

where $\alpha_{kld}$ and $\alpha_{acm}$ are hyperparameters for corresponding loss terms.

## 6 Experiments

In this section, we empirically evaluate our method for robustness transfer. We primarily show the effectiveness of our method under three settings: 1) Robustness Transfer in Section 6.1 to demonstrate that our method can transfer robustness from a pre-trained teacher to student without generating adversarial examples; 2) Robustness transfer from larger to smaller models in Section 6.2 to show its effectiveness under distillation settings; and 3) robustness transfer under transfer learning settings, e.g., from one dataset to another dataset and robustness with smaller dataset sizes in Section 6.3.

**Evaluation of Robustness**: We evaluate MixACM and other methods against standard FGSM and PGD-k as well as the strongest auto-attack by Croce and Hein [19]: a parameter-free adversarial attack designed to give a reliable evaluation of robustness. It is an ensemble of four white and black-box attacks. Apart from these, we also report robustness of our robust models on auto pgd-ce [19], auto pgd-dlr [19], FAB [18] and query-based black-box attack called Square attack [2]. For PGD-k, we use the standard value of step size $(2/255)$, and evaluated the model with an increasing number of iterations $(k)$, including an extreme version: PGD-500.

**Pre-Trained Robust Models**: For CIFAR experiments, we use WideResNet-28-10 and WideResNet-34-20 [90] provided by Gowal et al. [28]. For ImageNet experiments, we use adversarially trained ResNet-50 provided by Salman et al. [63].

**Training Details**: For CIFAR experiments, we use WideResNet-34-10 [90] as student model following [16]. All student models for our proposed method are trained without adversarial perturbations.

Table 2: **Comparison on ImageNet.** Results of robustness transfer for large-scale ImageNet dataset with ResNet-50 with two different training settings. Our method has significantly better robustness while also achieving comparable or better clean accuracy although it does not require any additional back-propagation steps.

| Method | Train $\epsilon$ | Acc. | PGD10 | PGD50 | PGD100 | Train $\epsilon$ | Acc. | PGD10 | PGD50 | PGD100 |
|---|---|---|---|---|---|---|---|---|---|---|
| | | Test $\epsilon = 2/255$ | | | | | Test $\epsilon = 4/255$ | | | |
| Free-AT [66] | 2 | **64.45** | 43.52 | 43.39 | 43.40 | 4 | 60.21 | 32.77 | 31.88 | 31.82 |
| Fast-AT [79] | 2 | 60.90 | - | 43.46 | - | 4 | 55.45 | - | 30.28 | - |
| Ours | 0 | 59.64 | 46.03 | 45.96 | 45.92 | 0 | 59.64 | 32.70 | 31.51 | 31.39 |
| Ours+RA | 0 | 62.05 | **48.54** | **48.45** | **48.44** | 0 | **62.05** | **34.93** | **33.71** | **33.63** |

Table 3: **Distillation Results.** Performance for robustness distillation on CIFAR-10 and CIFAR-100 datasets. The teacher model is WideResNet-28-10 and student model is WideResNet-16-10. Robustness is reported for diverse set of attacks of smagnitude $\epsilon = 8/255$.

| Dataset | Method | Acc. | FGSM | PGD 20 | PGD 500 | APGD CE | APGD DLR | FAB | Square | Auto Attack |
|---|---|---|---|---|---|---|---|---|---|---|
| CIFAR-10 | Natural | 93.79 | 5.34 | 0 | 0 | 0 | 0 | 0 | 0 | 0 |
| | PGD7-AT | 83.90 | 53.07 | 47.55 | 47.36 | 47.28 | 44.52 | 44.89 | 53.26 | 44.51 |
| | RKD | 88.60 | 27.37 | 0.12 | 0 | 0 | 0 | 0 | 7.31 | 0 |
| | Ours w/o mixup | 90.47 | 56.98 | 40.69 | 39.69 | 39.24 | 36.46 | 37.21 | 56.30 | 36.42 |
| | Ours | 89.93 | 59.48 | 48.09 | 47.51 | 47.31 | 43.73 | 44.46 | 60.20 | 43.64 |
| CIFAR-100 | Natural | 76.91 | 3.17 | 0 | 0 | 0 | 0 | 0 | 0 | 0 |
| | PGD7-AT | 60.29 | 28.37 | 24.94 | 24.75 | 24.71 | 21.94 | 22.16 | 27.97 | 21.94 |
| | RKD | 67.19 | 10.27 | 0.23 | 0.16 | 0.04 | 0 | 0 | 3.73 | 0 |
| | Ours w/o mixup | 60.81 | 30.13 | 25.38 | 25.12 | 24.93 | 19.51 | 19.83 | 29.74 | 19.50 |
| | Ours | 59.35 | 31.89 | 28.24 | 28.20 | 27.96 | 22.02 | 22.26 | 31.60 | 22.02 |

We trained them for 200 epochs, using batch size of 128, a learning rate of 0.1, cosine learning rate scheduler [46], momentum optimizer with weight decay of 0.0005. For our loss, we use $\alpha_{acm} = 5000$. For KD loss, we use temperature value of $\gamma = 10$ and $\alpha_{kld} = 0.95$ and the value for mixup coefficient is $\alpha_{mixup} = 1$ whereas $\lambda \sim Beta(\alpha_{mixup}, \alpha_{mixup})$ following [95]. ImageNet models are trained for 120 epochs. Additional training details are in the supplementary material.

## 6.1 Transferring Robustness without Adversarial Examples

**CIFAR-10**: To show the effectiveness of our method for robustness transfer and compare it with state-of-the-art, we follow settings of [16]: WideResNet-34-10 as a student on CIFAR-10. We compare our method with two kinds of defense methods: general variants of adversarial training and three robustness transfer methods. For our method, the average $\pm$ variance of five runs for our method are shown in Table 1. The robustness of our method is comparable to most state-of-the-arts adversarial training methods while still maintaining significantly higher clean accuracy.

**ImageNet**: We also consider high-resolution, large-scale ImageNet dataset [22]. This dataset is significantly more challenging for robustness [84, 83]. The results of our method for ResNet-50 student are shown in Table 2. For our method, we show two results: one with only mixup augmentation and one with mixup and random augmentation [20]. We observed that addition of random augmentation further improves the results of our method. Our method outperforms both Free [66] and Fast [79] AT in accuracy and robustness, significantly.

## 6.2 Robustness Distillation for Model Compression

A benefit of our method is its ability to distill robustness across models. To show this, we conduct two sets of experiments. First, we show robustness distillation from WideResNet-28-10 to WideResNet-16-10 for CIFAR-10 and CIFAR-100 datasets. The results are shown in Table 3. To see the effect of robustness transfer across different models, we also conduct an experiment where we changed the size of the model in terms of depth, the number of channels, or both, e.g., WideResNet-28-10 to WideResNet-16-5. The results are shown in Table 4. Our method works well for both larger-to-smaller

Table 4: **Model Compression**. Comparison of our method with PGD-7 Adversarial Training for accuracy and robustness on various sizes of models. The robustness is reported for $\ell_\infty$ PGD-20 attack of magnitude $\epsilon = 8/255$. Our method works well for robustness transfer from larger to smaller as well as, for smaller to larger models.

| Distill Type | Teacher | Student | Size Ratio | PGD7-AT Acc. | PGD7-AT Rob. | Ours Acc. | Ours Rob. |
|---|---|---|---|---|---|---|---|
| Depth | WRN-28-10 | WRN-16-10 | 46.92% | 83.90 | 47.55 | 89.93 | 48.09 |
| Channel | WRN-28-10 | WRN-28-5 | 25.04% | 83.90 | 47.74 | 90.26 | 48.75 |
| Channel | WRN-34-20 | WRN-34-10 | 25.01% | 84.30 | 49.72 | 87.26 | 52.83 |
| Depth & Channel | WRN-28-10 | WRN-16-5 | 11.76% | 83.51 | 48.11 | 88.96 | 36.95 |
| Depth & Channel | WRN-34-20 | WRN-16-10 | 9.227% | 83.90 | 47.55 | 86.31 | 47.18 |
| Tiny network | WRN-28-10 | WRN-10-1 | 0.21% | 67.11 | 38.85 | 66.65 | 1.15 |
| Tiny network | WRN-34-20 | WRN-10-1 | 0.04% | 67.11 | 38.85 | 63.19 | 3.75 |
| Same network | WRN-28-10 | WRN-28-10 | 100% | 84.17 | 49.23 | 90.48 | 54.05 |
| Same network | WRN-34-20 | WRN-34-20 | 100% | 84.20 | 49.11 | 87.17 | 53.60 |
| Up Transfer | WRN-28-10 | WRN-34-20 | 505.85% | 84.20 | 49.11 | 90.67 | 58.36 |

Figure 3: Robustness Transfer with Smaller Data. Figure shows robustness and accuracy (%) for the student model trained with different fractions of CIFAR-10 dataset. Our method achieves significantly better clean accuracy and robustness compared with adversarial training. Dotted blue line shows the performance of adversarial training with full data.

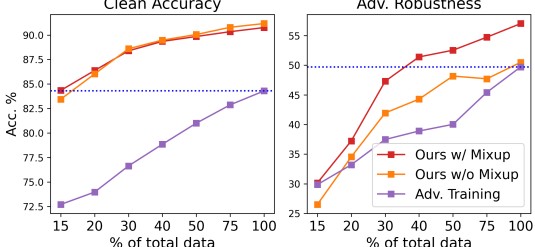

Table 5: Robustness Transfer from ImageNet to CIFAR-100. Teacher model is adversarially trained on ImageNet and student is trained on CIFAR-100. Our method achieves significantly better clean accuracy and better robustness compared with both adversarial training and IGAM [16]. The robustness is reported according to [16] for a fair comparison. Reported results for our method are mean of five repetitions.

| Defense | Acc. | FGSM | PGD-k 5 | PGD-k 10 | PGD-k 20 |
|---|---|---|---|---|---|
| Natural [16] | 78.70 | 7.95 | 0.13 | 0.03 | 0 |
| PGD7-AT [16] | 60.40 | 29.10 | 29.30 | 24.30 | 23.50 |
| IGAM [16] | 62.39 | 34.31 | 29.59 | 24.05 | 21.74 |
| Ours w/o Mixup | **68.91** | 29.09 | 28.65 | 21.70 | 20.27 |
| Ours | 65.69 | 32.09 | **32.04** | **25.58** | **24.14** |

as well as smaller-to-larger models. However, our method does not perform well when both depth and number of channels are decreased dramatically, e.g., from WideResNet-34-20 to WideResNet-10-1 (student's size is 0.04% of teacher's size). To elaborate further, we discuss this in the supplementary material.

## 6.3 Robust Transfer Learning

To further show the effectiveness of our method, we conduct two sets of experiments under transfer learning settings: transfer learning from one dataset to another dataset and transfer learning with fewer data samples. For first experiment, we follow Hendrycks et al. [32]'s setting: our teacher model is a WideResNet-28-10 trained on ImageNet of size $32 \times 32$ and student is a WideResNet-34-10. The results for the CIFAR-100 dataset under these settings are shown in Table 5.

For the second part, we show the effectiveness of our method with fewer data points. To this end, we randomly choose a smaller portion of CIFAR-10 and conducted experiments on this smaller portion of data with our method as well with PGD-7 adversarial training. The results are shown in Figure 3. Our method outperforms adversarial training significantly for both robustness and clean accuracy. Using just 30% of the data, our method can match the robustness and clean accuracy of a robust model trained on all the data.

Table 6: **Ablation Studies**: (a) Impact of different components of our proposed distillation method. We use WideResNet-34-10 as student following Table 1 in the main text. (b) Impact of of using different intermediate features on clean accuracy and robustness of student.

(a)

|  | Std. Setting | w/o KD | w/o Mixup | Only ACM | Only KD |
|---|---|---|---|---|---|
| Mapping ($g_c$) | ✓ | ✓ | ✓ | ✓ | |
| Soft Labels | ✓ | | ✓ | | ✓ |
| Mixup | ✓ | ✓ | | | |
| ACM Loss | ✓ | ✓ | ✓ | ✓ | |
| Accuracy | 90.76 | 92.50 | 91.17 | 92.87 | 89.38 |
| Robustness | 56.65 | 52.29 | 49.53 | 48.38 | 0.21 |

(b)

| Features Used | Acc. | Rob. |
|---|---|---|
| Low-Level (2) | 86.36 | 17.24 |
| Mid-Level (3) | 88.30 | 39.37 |
| High-Level (4) | 91.18 | 33.13 |
| Low+Mid Level (2+3) | 88.15 | 41.92 |
| Mid+High Level (3+4) | 90.69 | 52.79 |
| First Layer Only (1) | 86.00 | 0.31 |
| No First Layer (2+3+4) | 90.69 | 56.15 |
| All Features | 90.76 | 56.65 |

## 6.4 Ablation Studies

**Effect of Individual Components:** To see the effect of individual components of our proposed method, we perform an experiment where we try different settings. The results are shown in Table 6(a). ACM loss alone is sufficient to transfer significant robustness but adding mixup and soft labels improves the transferability. However, the improvement in robustness comes at the cost of clean accuracy. Also, note that only soft labels are not enough to transfer robustness as shown by KD Only column. This is in line with the observations of Goldblum et al. [25].

**Role of Intermediate Features**: To understand the role of low, mid, and high-level features, we performed experiments on CIFAR-10 by progressively changing blocks used for distillation. For this ablation study, we kept all the standard settings reported in the paper (supplementary material section A.1). Our correspondence of blocks and features is as follows: block 2: low-level features; block 3: mid-level features; block 4: high-level features. Please note that block 1 corresponds to the output of the first layer only. Therefore, we do not call it low-level features.

The results are reported in Table 6 (b). In summary, all level features (low, mid, high level) improve robustness and accuracy. However, mid-level features seem to be more critical for robustness and high-level features for accuracy. For more details, please refer to supplementary material.

**Other Ablation Studies**: We have conducted several additional ablation studies and discussed them in the supplementary material (Section A). We recapitulate the purpose of these ablation studies here for reference. First, we conducted a study to understand the effect of direct distillation vs. distillation via MixACM. Second, we performed an ablation study to see the effect of three hyperparameters used by our loss ($\alpha_{acm}, \alpha_{kld}, \gamma$). Third, we performed a study to see how different transforms ($g_c(.)$) effect the results. Fourth, to see the limitation of our method, we performed an experiment with progressively decreasing the size of student. Finally, we also compared activation maps of a normally trained student, teacher and model trained with our method.

## 7 Conclusion

In this paper, we have investigated robustness transfer from a theoretical and algorithmic perspective. The general principle behind most successful adversarial defense methods is the generation of adversarial examples during training. However, adversarial training is complicated as it requires more data, larger models, higher compute resources and degrades clean accuracy. We investigated whether we can have robustness more efficiently through robustness transfer. To this end, we first presented a theoretical result bounding adversarial loss of student model with the distance between robust teacher and student on mixup examples. We, then, proposed a novel robustness transfer method that is based on channel-wise analysis of robustness. Our method can achieve significant robustness against challenging and reliable auto-attack as well as other standard attacks while outperforming previous work in terms of clean accuracy. The proposed method also works well for the distillation of robustness to a smaller model, and for transfer learning with fewer data points.

**Acknowledgements**: Authors are thankful to Ferjad Naeem, Dr. Nauman, and anonymous reviewers for their valuable feedback.

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
