# Supplementary Material
# MixACM: Mixup-Based Robustness Transfer via Distillation of Activated Channel Maps

**Muhammad Awais** [1, 2]*[†] **Fengwei Zhou** [1]* **Chuanlong Xie** [1]* **Jiawei Li** [1],
**Sung-Ho Bae** [2]‡ **Zhenguo Li** [1]

[1] Huawei Noah's Ark Lab
[2] Department of Computer Science, Kyung-Hee University, South Korea
awais@khu.ac.kr, {zhoufengwei, xie.chuanlong, li.jiawei}@huawei.com,
shbae@khu.ac.kr, li.zhenguo@huawei.com

## Contents

## A   Experimental Results

### A.1   Additional Details of Experimental Setup

**Baselines:** For fair comparisons with other methods, we either use the best results reported in the paper or retrained models with optimal hyper-parameters described by the papers. Details of

---

*Equal Contribution.

†This work was carried out at Huawei Noah's Ark Lab. The webpage for the project is available at: awaisrauf.github.io/MixACM

‡Corresponding Author.

35th Conference on Neural Information Processing Systems (NeurIPS 2021).

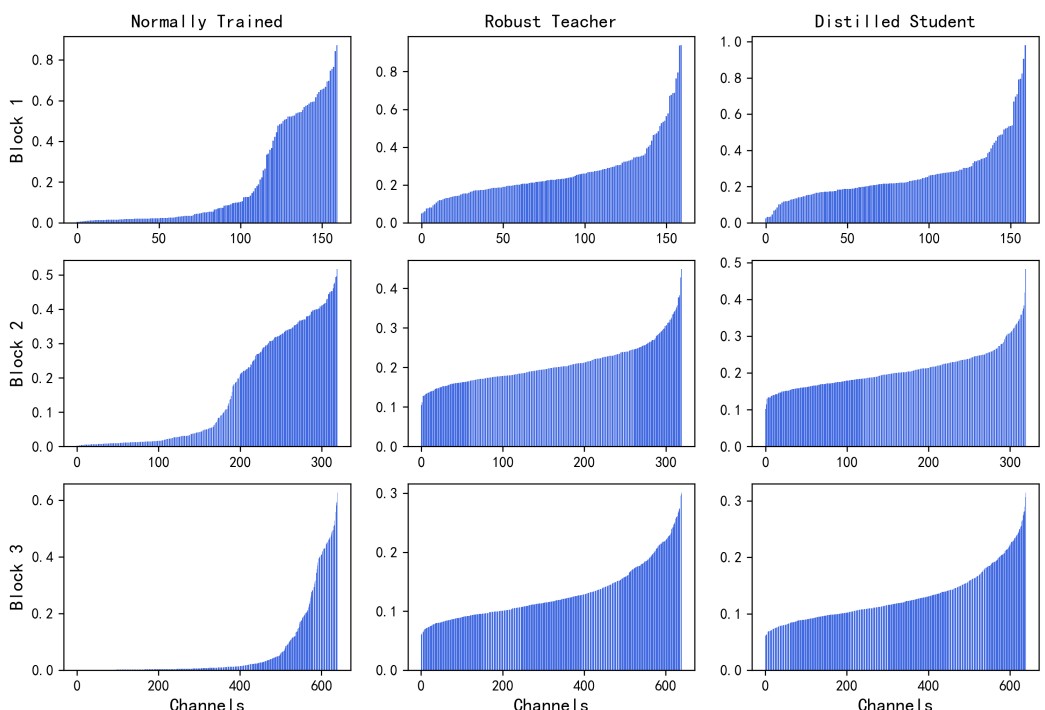

Figure 4: **Comparison of Activated Channel Maps.** Comparison of Activated Channel Maps of a normally trained model, robust teacher and student trained with the proposed method. The $x$-axis of figure represents channel number of a specific layer and $y$-axis represents ACM value. Our method makes ACM of student similar to the robust teacher.

comparison for all the experimental results in the main text are as follows. For Table 1, we take clean accuracy and auto-attack results from [2] and PGD-100 results are the best PGD attack reported results (with the same or similar setting as ours) taken from the respective papers. For Table 2, we take Shafahi et al. [11], Wong et al. [12]'s reported results and evaluated our model with the same settings of PGD attack. For Table 3, we train the same models with PGD7-AT [8], RKD [4] and our method. For PGD7-AT and RKD, we use optimal hyper-parameters reported in the papers. For Table 4, we train the same models with PGD7-AT [8], and our method and evaluated all models with the same settings of PGD attack. The size ratio is computed based on the number of trainable parameters of the student to the trainable parameters of teacher. The purpose of Table 5 is a comparison of our method with IGAM [1] under transfer learning settings. We used PGD7-AT and normal results reported by them. Unlike them, however, results of our method are mean of five repetitions. For Figure 3, we train the same models with the same proportion of data with PGD7-AT and our method.

**Adversarial Training:** We use standard PGD-7 Adversarial Training with the step size of $2/255$ and $\epsilon = 8/255$.

**Teacher Models:** We use four teachers in the paper. For most of our CIFAR experiments, we use a WideResNet-28-10 trained by Gowal et al. [5]. For some experiments, we also use WideResNet-34-20 trained on CIFAR-10 by Gowal et al. [5] and WideResNet-28-10 trained on tiny ImageNet trained by [6]. For ImageNet experiments, we use ResNet-50 provided by [10].

**Student Models:** Student models share all the architectural design of respective teachers e.g. if teacher model by Gowal et al. [5] uses Swish activation function, student models also uses Swish activation. For main CIFAR-10 experiments, we use WideResNet-34-10 following many relevant works like Zhang et al. [13], Chan et al. [1], Pang et al. [9], etc. We also use students with different widths (20, 10, 5, 1) and depths (34, 28, 22, 16, 10). For ImageNet, we use ResNet50 following [11, 12].

**Optimizer Setting:** For CIFAR experiments, we use SGD with Nesterov momentum 0.9, initial learning rate of 0.1, cosine annealing learning rate decay without restarts, weight decay of $5 \times 10^{-4}$

and a batch size of 128. For ImageNet, the model is trained for 120 epochs by SGD with a momentum 0.9, weight decay of $5 \times 10^{-5}$, batch size of 2048, initial learning rate of 0.8, and cosine learning rate decay. We also use a gradual warmup strategy that increases the learning rate from 0.16 to 0.8 linearly in the first 5 epochs.

**Augmentation:** All the experiments of our method use Mixup augmentation with coefficient 1 for CIFAR and 0.2 for ImageNet; unless mentioned otherwise. For CIFAR, we also use standard augmentation: randomly cropping a part of $32 \times 32$ from the padded image followed by a random horizontal flip provided by PyTorch. For ImageNet, we do two experiments: one with only mixup and one with mixup and random augmentation.

**Hyper-parameter Selection:** Our method adds a new hyper-parameter $\alpha_{acm}$. Two other hyper-parameters of our loss are $\alpha_{kld}$ and temperature $\gamma$. The selection process for them is detailed in Section A.4. For $\alpha_{acm}$, we use ablation study to find optimal range of $\alpha_{acm}$ and all of the other experiments are done with $\alpha_{acm} \in \{2000, 5000\}$ and best results are reported.

**Compute Infrastructure:** We train CIFAR models on one NVIDIA Tesla V100. For ImageNet, we train models in a distributed fashion using 32 GPUs in the cloud.

## A.2 Activated Channel Maps

Our method transfers robustness by matching Activated Channel Maps (ACMs) of robust teacher and student on natural examples. To get Activated Channel Maps, we first get an output of a model at a specific layer called activations $\mathcal{A}$. These activations are then passed through a mapping function to get activation maps: $a_i = g_c(\mathcal{A}_i)$ and normalized with the magnitude. This process is illustrated in Figure 5. The size of activated channel map is equal to number of channels in a layer e.g. if activation has a size of $\mathcal{A}_i \in \mathbb{R}^{C \times H \times W}$ then activated channel maps shape will be $a_i \in \mathbb{R}^{C \times 1 \times 1}$.

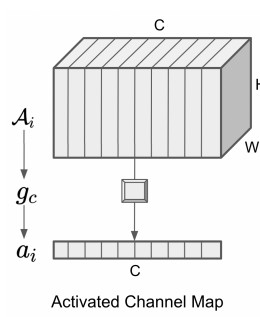

Figure 5: Mapping function $g_c$ is applied channel-wise to get the Activated Channel Maps (ACM).

We show Activated Channel Maps of three different blocks of a normally trained WideResNet-16-10, a robust teacher WideResNet-28-10 and a student WideResNet-16-10 trained with our method in Figure 4. The maps are generated on 500 natural examples of CIFAR-10 test set. The Figure shows an average of maps produced on all the examples and sorted in ascending order. The $x$-axis in the Figure represents channels and the $y$-axis represents the Activated Channel Map values. The distribution of Activated Channel Maps of our method is spectacularly similar to the robust teacher.

## A.3 Limitation of Proposed Method for Distillation

We show results for distillation with different settings in the paper. To see the limits of our distillation method, we perform an experiment where we progressively reduce the number of channels and layers of the student. The results are shown in Figure 6. For individual values and comparison with PGD7-AT, please see Table 10.

We observe that our method works well for a compression ratio of $> 0.1$ (student has $0.1\times$ teacher's trainable parameters). We also observe that robustness transfer deteriorates significantly if depth or width is reduced significantly (e.g., depth reduced from 28 to 10; width reduced from from 10 to 1). The degradation based on width can be attributed to the transformation function and we expect that other functions may work better. The depth degradation may be due to the representational capability of the student model. We leave further investigation of this for future work.

## A.4 Ablation Studies

### A.4.1 Effect of Individual Components

To see the effect of individual components of our proposed method, we perform an experiment where we switch different components. The results are shown in Table 6.

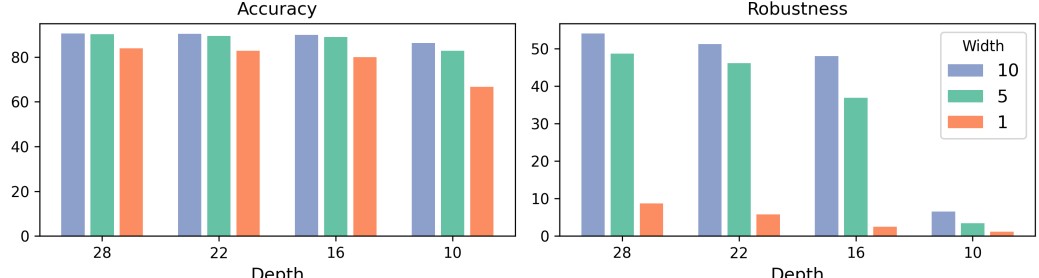

(a) The figure shows effect of decreasing number of layers and number of channels on accuracy and robustness for our method. The teacher model is a robust WideResNet-28-10 and student models are WideResNet-*Depth-Width*.

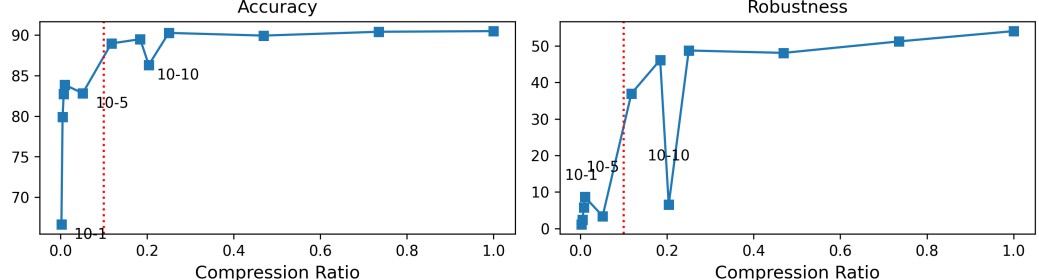

(b) The figure shows compression ration vs accuracy and robustness for our method. The dotted red line represents compression ratio of 0.1. Our method works in transferring robustness beyond this point except for dips caused by WideResNet of depth 10.

Figure 6: **Limits of Distillation.** Plots show limits of our method in distilling a large teacher model's robustness to a small student model without adversarial training.

Table 6: **Impact of Individual Components.** Impact of different components of our proposed distillation method. We use WideResNet-34-10 as student following Table 1 in the main text.

|  | Std. Setting | w/o KD | w/o Mixup | Only ACM | Only KD |
|---|---|---|---|---|---|
| Mapping Function ($g_c$) | ✓ | ✓ | ✓ | ✓ | |
| Teacher Soft Labels | ✓ | | ✓ | | ✓ |
| Mixup | ✓ | ✓ | | | |
| ACM Loss | ✓ | ✓ | ✓ | ✓ | |
| Accuracy | 90.76 | 92.50 | 91.17 | 92.87 | 89.38 |
| Robustness | 56.65 | 52.29 | 49.53 | 48.38 | 0.21 |

In summary, ACM loss alone can transfer significant robustness from the teacher; the addition of soft labels and mixup further improves this transfer. Specifically, robustness with only ACM loss is 48.38%, the addition of soft-labels improves it to 49.53%, the addition of mixup improves it to 52.29%, and the addition of both of these components make final robustness to 56.65%. Also, note that only soft labels are not enough to transfer robustness in this case, as shown by KD Only column. This is in line with the observations of Goldblum et al. [4].

### A.4.2 Role of Intermediate Features

To understand the role of low, mid, and high-level features, we performed experiments on CIFAR-10 by progressively changing blocks used for distillation. For this ablation study, we kept all the standard settings reported in the Section A.1. Our correspondence of blocks and features is as follows: block 2: low-level features; block 3: mid-level features; block 4: high-level features. Please note that block 1 corresponds to the output of the first layer only. Therefore, we do not call it low-level features. The results are shown in the Table 7.

| Features Used | Accuracy | Robustness |
|---|---|---|
| Low-Level (2) | 86.36 | 17.24 |
| Mid-Level (3) | 88.30 | 39.37 |
| High-Level (4) | 91.18 | 33.13 |
| Low+Mid Level (2+3) | 88.15 | 41.92 |
| Mid+High Level (3+4) | 90.69 | 52.79 |
| First Layer Only (1) | 86.00 | 0.31 |
| No First Layer (2+3+4) | 90.69 | 56.15 |
| All Features | 90.76 | 56.65 |

Table 7: Impact of of using different intermediate features on clean accuracy and robustness of student.

As shown in the Table 7, mid-level features play a more important role in robustness transfer and high-level features play a crucial role in accuracy transfer. The robustness of the student is 39.37% when we only use mid-level features, but it decreases to 33.13% when high-level features are utilized alone. On the other hand, clean accuracy improves when we only use high-level features: 88.30% with mid-level compared with 91.18% with high-level features. In addition, a combination of mid and high-level features is enough to get close to optimal robustness and accuracy. But the addition of low-level features improves robustness even further.

Apart from these low, mid, and high-level features, we also used the output of the first layer in the proposed loss function. Our experiments above show that the improvement brought by first layer distillation is relatively small. Specifically, the addition of the first layer in the above-mentioned experiments brings $\leq 1\%$ improvement for robustness.

In summary, all level features (low, mid, high level) improve robustness and accuracy. However, mid-level features seem to be more critical for robustness and high-level features for accuracy.

### A.4.3 Comparison of Losses

The purpose of ACM loss is to match activated channel maps of teacher and student. It is possible to distill robustness by directly matching intermediate features of teacher and student. However, this direct way of distillation overlooks differences between the teacher and the student such as structure, number of channels, size of activations, how and on what data teacher is trained, etc.

To see the effect of directly distilling the intermediate features, we also have conducted an ablation study comparing direct distillation ($\ell_2$-loss) with ACM-based distillation while progressively increasing differences between the teacher and the student. We have kept all the standard settings (Section A.1) and used similar settings for direct distillation for a fair comparison.

The results are reported in Table 8. When teacher and student are similar, ACM performs slightly better than direct distillation (56.65% vs. 56.12%). However, when the number of channels of teacher and student is different, the performance gap increases (48.75% vs. 44.95%). This gap increases further when both channels and the number of layers are different (47.18% vs. 41.90%). A similar gap is also visible in terms of clean accuracy for all these cases.

To further explore the effect of this difference, we also performed one experiment under transfer learning settings for CIFAR-100 (Table 5 in the paper). Here, the teacher is trained on a different dataset (ImageNet), so the difference between the two models is larger. The performance gap is also wider. ACM outperforms direct distillation significantly (clean accuracy: 65.69% vs. 57.86% and robustness: 24.14% vs. 16.20%).

### A.4.4 Effect of $\alpha_{acm}$

The only extra hyper-parameters introduced by our algorithm is the weight of ACM loss ($\alpha_{acm}$). To see the effect of $\alpha_{acm}$, we perform an ablation experiment with WideResNet-34-10 as student. To avoid any confounding effect of other factors, we use only ACM loss in this experiment. The results are shown in Figure 7(a).

In summary, the clean accuracy of the model is less sensitive to $\alpha_{acm}$ compared with robustness. For instance when we vary the values of $\alpha_{acm}$ from 100000 to 100, the clean accuracy changes from 94

| Method | Teacher | Student | Accuracy | Robustness |
|---|---|---|---|---|
| Distillation for CIFAR-10 | | | | |
| Full | WRN28-10 | WRN34-10 | 89.98 | 56.12 |
| MixACM | WRN28-10 | WRN34-10 | **90.76** | **56.65** |
| Full | WRN28-10 | WRN28-5 | 88.46 | 44.95 |
| MixACM | WRN28-10 | WRN28-5 | **90.26** | **48.75** |
| Full | WRN-34-20 | WRN16-10 | 84.27 | 41.9 |
| MixACM. | WRN-34-20 | WRN16-10 | **86.31** | **47.18** |
| Transfer Learning for CIFAR-100 | | | | |
| Direct | WRN28-10 | WRN34-10 | 57.86 | 16.20 |
| ACM | WRN28-10 | WRN34-10 | 65.69 | 24.14 |

Table 8: Effect of using direct loss vs. MixACM loss on distillation and transfer learning.

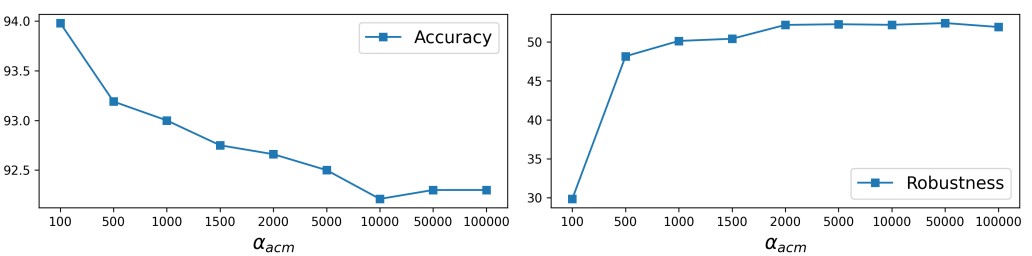

(a) Effect of $\alpha_{acm}$ on accuracy and robustness.

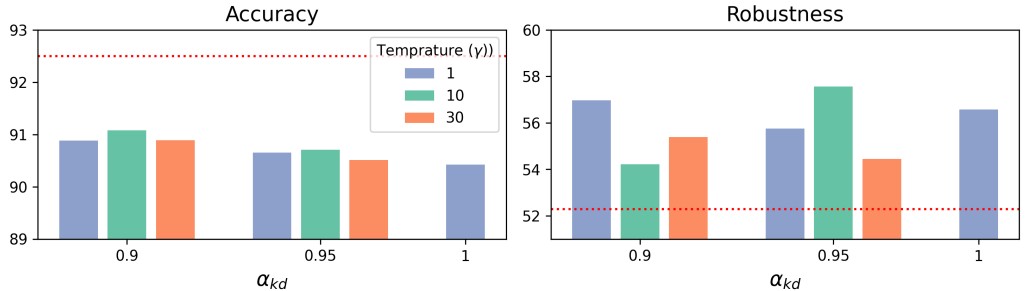

(b) Effect of $\alpha_{kld}$ and temperature $\gamma$ on accuracy and robustness. The dotted red line represents training without soft labels (i.e., $\alpha_{kld} = 0$).

Figure 7: **Hyper-parameter Selection.** Comparison of three different hyper-parameters on accuracy and robustness.

to 92 while robustness changes from 50 to 30. The value of $\alpha_{acm}$ also acts as a trade-off between clean accuracy and robustness.

### A.4.5 Effect of $\alpha_{kld}$ and temperature $\gamma$

Our proposed method uses soft labels with KLD loss [7]. This loss has two hyper-parameters: $\alpha_{kld}$ and temperature $\gamma$. We select best values of these hyper-parameters with an experiment where we set $\alpha_{acm} = 5000$ and tried three different settings of these two parameters. The results are reported in Figure 7(b). The student is a WideResNet-34-10. We select best values: $\alpha_{kld} = 0.95, \gamma = 10$. We use these hyper-parameter values for all of our experiments.

Table 9: Effect of different transformation functions for robustness transfer from robust WideResNet-28-10 to WideResNet-28-5.

| Teacher Transform | Student Transform | Accuracy | Robustness |
|---|---|---|---|
| None | Adaptive Max Pool | 89.92 | 46.41 |
| None | Adaptive Avg Pool | 90.04 | 46.27 |
| None | Affine | 88.86 | 30.27 |
| Adaptive Max Pool | None | 90.26 | 48.75 |
| Adaptive Avg Pool | None | 89.99 | 42.01 |
| Affine | None | 88.39 | 23.51 |

Table 10: **Distillation with Different Size Ratios.** The table shows result of distillation from a large teacher to a progressively small student. Size ratio is ratio of number of trainable parameters in student to teacher. Our method works well to reduce teacher size $10\times$. However, significant compression of depth and width causes a sharp deterioration of performance.

| Teacher | Student | Size Ratio | PGD7-AT | | Ours | |
|---|---|---|---|---|---|---|
| | | | Acc. | Rob. | Acc. | Rob. |
| WRN 28-10 | WRN 28-10 | 100% | 84.17 | 49.23 | 90.48 | 54.05 |
| | WRN 28-5 | 25.04% | 83.90 | 47.74 | 90.26 | 48.75 |
| | WRN 28-1 | 1.01% | 78.89 | 48.27 | 83.86 | 8.67 |
| | WRN 22-10 | 73.5% | 84.26 | 49.29 | 90.40 | 51.23 |
| | WRN 22-5 | 18.40% | 83.73 | 48.93 | 89.49 | 46.16 |
| | WRN 22-1 | 0.75% | 78.36 | 47.57 | 82.74 | 5.81 |
| | WRN 16-10 | 46.92% | 83.90 | 47.55 | 89.93 | 48.09 |
| | WRN 16-5 | 11.76% | 83.51 | 48.11 | 88.96 | 36.95 |
| | WRN 16-1 | 0.48% | 75.69 | 44.46 | 79.90 | 2.45 |
| | WRN 10-10 | 20.38% | 78.59 | 44.41 | 86.31 | 6.54 |
| | WRN 10-5 | 5.12% | 78.16 | 44.04 | 82.82 | 3.41 |
| | WRN 10-1 | 0.21% | 67.11 | 38.85 | 66.65 | 1.15 |

### A.4.6 Comparison of Transforms

Our method requires a transformation function applied on Activated Channel Maps if teacher and student have different number of channels. For this purpose, we compare performance of three different transformation functions: a fully connected layer, an adaptive average pool function or adaptive max pool function. We applied these functions under two settings: applied on teacher's activated channel maps or applied on student's activated channel maps. The results are shown in Table 9.

## B   Theoretical Results

This appendix has full proofs for results in Section 4. The order is as follows. We first prove that the population adversarial error can be bounded above by the sum of an empirical adversarial loss and a distillation loss. Then we consider a binary classification task with the logistic loss function and prove that the distillation loss can be approximated by the mixup augmentation based loss.

### B.1   Proof of Theorem 4.1

**Theorem.** *We consider task of mapping input $x \in \mathcal{X} \subseteq \mathbb{R}^d$ to label $y \in \mathcal{Y} = \{1, 2, \ldots, K\}$. Denote the training data as $\mathcal{D} = \{(x_1, y_1), ..., (x_n, y_n)\}$, where each data point is sampled from a ground truth distribution $P$. The goal is to learn a classifier $f : \mathcal{X} \to \mathbb{R}^K$ from a hypothetical space $\mathcal{F}$. Let $L(f(x), y) = 1 - \phi_\gamma(f(x), y)$ be the loss function, where $\gamma > 0$ is the temperature and*

$$\phi_\gamma(f(x), y) = \frac{\exp(f(x)_y/\gamma)}{\sum_{k=1}^K \exp(f(x)_k/\gamma)}.$$

*Suppose a classifier $f$ is decomposed into $g \circ h$, where $h \in \mathcal{H}$ is a feature extractor and $g \in \mathcal{G}$ stands for the top model. We considered supervision of a teacher feature extractor $h^T(x)$ trained on same or similar dataset. Then, with probability $1 - \delta$, for any $f \in \mathcal{F}$,*

$$\mathbb{E}\Big[\max_{\|\delta\|_\infty \leq \epsilon} \mathbb{I}\{\hat{y}(x+\delta) \neq y\}\Big] \leq 2\tilde{\mathcal{L}}(f, \mathcal{D}) + \frac{8}{\gamma}\tilde{\mathcal{L}}_{dis}(f, \mathcal{D}) + 4\mathfrak{R}_S(\Psi_T) + \frac{16}{n} + 6\sqrt{\frac{2/\delta}{2n}}, \quad (1)$$

*where $\mathfrak{R}_S$ is the Rademacher complexity, $\Psi_T = \big\{\max_{\|\delta\|_\infty < \epsilon} L(g \circ h^T(x+\delta), y) : \ g \in \mathcal{G}\big\}$ and*

$$\tilde{\mathcal{L}}_{dis}(f, \mathcal{D}) = \frac{1}{n}\inf_{g \in \mathcal{G}}\sum_{i=1}^{n}\max_{\|\delta\|_\infty < \epsilon}\big|f(x_i+\delta) - g \circ h^T(x_i+\delta)\big|.$$

**Proof:** Denote the adversarial loss with $\epsilon$-bounded $\ell_\infty$ adversarial attacks as

$$\tilde{L}(f(x), y) = \max_{\|\delta\|_\infty < \epsilon} L\big(f(x+\delta), y\big).$$

Let $j \in \{1, \dots, \log_2(n)\}$ and $\tau_j = 2^{2-j}$. Here, for simplicity, we assume $\log_2(n)$ is a positive integer. Then, we denote the following classes of functions $\Psi_T = \{\tilde{L}(g \circ h^T(x), y) : \ g \in \mathcal{G}\}$ and

$$\Psi_j = \{\tilde{L}(f(x), y) : \ \inf_{g \in \mathcal{G}}\frac{1}{n}\sum_{i=1}^{n}|\tilde{L}(f(x_i), y_i) - \tilde{L}(g \circ h^T(x_i), y_i)| \leq \tau_j\}.$$

By the classical generalization bound with the Rademacher complexity, with probability at least $1 - \delta$, for any $\tilde{L} \in \Psi_j$,

$$\mathbb{E}\big[\tilde{L}(f(x), y)\big] \ \leq \ \frac{1}{n}\sum_{i=1}^{n}\tilde{L}(f(x_i), y_i) + 2\mathfrak{R}_S(\Psi_j) + 3\sqrt{\frac{2/\delta}{2n}}, \quad (2)$$

where

$$\mathfrak{R}_S(\Psi_j) \ = \ \frac{1}{n}\mathbb{E}_\varepsilon\Big[\sup_{\tilde{L}(f) \in \Psi_j}\sum_{i=1}^{n}\varepsilon_i\tilde{L}\big(f(x_i), y_i\big)\Big].$$

Furthermore, we have

$$\begin{aligned}
\mathfrak{R}_S(\Psi_j) \ &= \ \frac{1}{n}\mathbb{E}_\varepsilon\Big[\sup_{\tilde{L} \in \Psi_j}\inf_{g \in \mathcal{G}}\sum_{i=1}^{n}\varepsilon_i\tilde{L}\big(f(x_i), y_i\big)\Big] \\
&= \ \frac{1}{n}\mathbb{E}_\varepsilon\Big[\sup_{\tilde{L}(f) \in \Psi_j}\inf_{g \in \mathcal{G}}\sum_{i=1}^{n}\varepsilon_i\big(\tilde{L}(f(x_i), y_i) - \tilde{L}(g \circ h^T(x_i), y_i) + \tilde{L}(g \circ h^T(x_i), y_i)\big)\Big] \\
&\leq \ \frac{1}{n}\mathbb{E}_\varepsilon\Big[\sup_{\tilde{L}(f) \in \Psi_j}\inf_{g \in \mathcal{G}}\max_i|\varepsilon_i|\sum_{i=1}^{n}\big|\tilde{L}(f(x_i), y_i) - \tilde{L}(g \circ h^T(x_i), y_i)\big|\Big] \\
&\quad + \frac{1}{n}\mathbb{E}_\varepsilon\Big[\sup_{g \in \mathcal{G}}\sum_{i=1}^{n}\varepsilon_i\tilde{L}(g \circ h^T(x_i), y_i)\Big] \\
&\leq \ \tau_j + \mathfrak{R}_S(\Psi_T).
\end{aligned}$$

Plugging the upper bound of $\mathfrak{R}_S(\Psi_j)$ into (2), we have with probability at least $1 - \delta$,

$$\mathbb{E}\big[\tilde{L}(f(x), y)\big] \ \leq \ \frac{1}{n}\sum_{i=1}^{n}\tilde{L}(f(x_i), y_i) + 2\tau_j + 2\mathfrak{R}_S(\Psi_T) + 3\sqrt{\frac{2/\delta}{2n}}. \quad (3)$$

Then the inequality (3) holds for all $f \in \mathcal{F}$ and $j$ with probability at least $1 - \log_2(n)\delta$. Given $f$, then for any $g \in \mathcal{G}$,

$$\frac{1}{n}\sum_{i=1}^{n}\big|\tilde{L}(f(x_i), y_i) - \tilde{L}(g \circ h^T(x_i), y_i)\big| \leq 2.$$

This implies that for any given $f$, there exists $j \in \{1, \ldots, \log_2(n)\}$ such that $\tilde{L}(f(x), y) \in \Psi_j$. We select the smallest $j$ that satisfies

$$\tau_j \geq \frac{1}{n} \inf_{g \in \mathcal{G}} \sum_{i=1}^{n} \left| \tilde{L}(f(x_i), y_i) - \tilde{L}(g \circ h^T(x_i), y_i) \right| \geq \frac{1}{2} \tau_j - 2^{1 - \log_2(n)}.$$

Then

$$\tau_j \leq \frac{4}{n} + \frac{2}{n} \inf_{g \in \mathcal{G}} \sum_{i=1}^{n} \left| \tilde{L}(f(x_i), y_i) - \tilde{L}(g \circ h^T(x_i), y_i) \right|.$$

Furthermore, the inequality (3) can be rewritten as

$$\begin{aligned}
\mathbb{E}\left[ \tilde{L}(f(x), y) \right] \quad \leq \quad & \frac{1}{n} \sum_{i=1}^{n} \tilde{L}(f(x_i), y_i) + \frac{8}{n} \\
& + \frac{4}{n} \inf_{g \in \mathcal{G}} \sum_{i=1}^{n} \left| \tilde{L}(f(x_i), y_i) - \tilde{L}(g \circ h^T(x_i), y_i) \right| \\
& + 2\mathfrak{R}_S(\Psi_T) + 3\sqrt{\frac{2/\delta}{2n}},
\end{aligned} \tag{4}$$

with probability at least $1 - \delta$. In addition,

$$\begin{aligned}
\tilde{L}(f(x), y) \quad & \leq \quad \max_{\|\delta\|_\infty < \epsilon} \left| L(f(x), y) - L(g \circ h^T(x), y) \right| + \tilde{L}(g \circ h^T(x), y), \\
\tilde{L}(g \circ h^T(x), y) \quad & \leq \quad \max_{\|\delta\|_\infty < \epsilon} \left| L(f(x), y) - L(g \circ h^T(x), y) \right| + \tilde{L}(f(x), y).
\end{aligned}$$

So we have

$$\begin{aligned}
\mathbb{E}\left[ \tilde{L}(f(x), y) \right] \quad \leq \quad & \frac{1}{n} \sum_{i=1}^{n} \tilde{L}(f(x_i), y_i) + \frac{8}{n} \\
& + \frac{4}{n} \inf_{g \in \mathcal{G}} \sum_{i=1}^{n} \max_{\|\delta\|_\infty < \epsilon} \left| L(f(x_i + \delta), y_i) - L(g \circ h^T(x_i + \delta), y_i) \right| \\
& + 2\mathfrak{R}_S(\Psi_T) + 3\sqrt{\frac{2/\delta}{2n}},
\end{aligned}$$

Next we show the relationship between the adversarial accuracy and $\mathbb{E}\left[ \tilde{L}(f(x), y) \right]$. The adversarial loss function $\tilde{L}$ can be rewritten as

$$\begin{aligned}
\tilde{L}(f(x), y) \quad = \quad & \max_{\|\delta\|_\infty < \epsilon} \frac{\sum_{k \neq y} \exp\left( f(x + \delta)_k / \gamma \right)}{\sum_{k \in [K]} \exp\left( f(x + \delta)_k / \gamma \right)} \\
= \quad & \max_{\|\delta\|_\infty < \epsilon} \frac{1}{1 + \frac{\exp\left( f(x+\delta)_y / \gamma \right)}{\sum_{k \neq y} \exp\left( f(x+\delta)_k / \gamma \right)}} \\
= \quad & \max_{\|\delta\|_\infty < \epsilon} \frac{1}{1 + \exp\left( f(x + \delta)_y / \gamma - \ln(\sum_{k \neq y} \exp(f(x + \delta)_k / \gamma)) \right)} \\
= \quad & \max_{\|\delta\|_\infty < \epsilon} \sigma\left( -\frac{f(x + \delta)_y}{\gamma} + \ln\left( \sum_{k \neq y} \exp(\frac{f(x + \delta)_k}{\gamma}) \right) \right),
\end{aligned} \tag{5}$$

where $\sigma$ is the sigmoid function. Notice that $\sigma$ is a monotonically increasing function and

$$\ln\left( \sum_{k \neq y} \exp(\frac{f(x + \delta)_k}{\gamma}) \right) \geq \max_{k \neq y} \frac{f(x + \delta)_k}{\gamma}.$$

Then we have

$$\tilde{L}(f(x), y) \geq \max_{\|\delta\|_\infty < \epsilon} \sigma\left( -\frac{f(x + \delta)_y}{\gamma} + \max_{k \neq y} \frac{f(x + \delta)_k}{\gamma} \right) \tag{6}$$

If there exists $\delta$ such that $f(x + \delta)_y \leq \max_{k \neq y} f(x + \delta)_k$, then

$$\max_{\|\delta\|_\infty < \epsilon} \sigma\Big( -\frac{f(x+\delta)_y}{\gamma} + \max_{k \neq y} \frac{f(x+\delta)_k}{\gamma} \Big) \geq \frac{1}{2} \max_{\|\delta\|_\infty < \epsilon} \mathbb{I}\big( f(x+\delta)_y \leq \max_{k \neq y} f(x+\delta)_k \big). \quad (7)$$

In contrast, if for any $\delta$, $f(x + \delta)_y > \max_{k \neq y} f(x + \delta)_k$, then

$$\sigma\Big( -\frac{f(x+\delta)_y}{\gamma} + \max_{k \neq y} \frac{f(x+\delta)_k}{\gamma} \Big) \geq \mathbb{I}\big( f(x+\delta)_y \leq \max_{k \neq y} f(x+\delta)_k \big). \quad (8)$$

Combining (7) and (8),

$$2\tilde{L}(f(x), y) \geq \max_{\|\delta\|_\infty < \epsilon} \mathbb{I}\big( f(x+\delta)_y \leq \max_{k \neq y} f(x+\delta)_k \big). \quad (9)$$

According to (4) and (9), we have

$$\mathbb{P}\big[ \{(x,y) : \exists \delta,\ \|\delta\|_\infty < \epsilon \text{ s.t. } \hat{y}(x+\delta) \neq y \} \big]$$
$$= \mathbb{E}\big[ \max_{\|\delta\|_\infty < \epsilon} \mathbb{I}\big( f(x+\delta)_y \leq \max_{k \neq y} f(x+\delta)_k \big) \big]$$
$$\leq 2\mathbb{E}\big[ \tilde{L}(f(x), y) \big]$$
$$\leq \frac{2}{n} \sum_{i=1}^n \tilde{L}(f(x_i), y_i) + \frac{16}{n}$$
$$+ \frac{8}{n} \inf_{g \in \mathcal{G}} \sum_{i=1}^n \max_{\|\delta\|_\infty < \epsilon} \big| L(f(x_i + \delta), y_i) - L(g \circ h^T(x_i + \delta), y_i) \big|$$
$$+ 4\mathfrak{R}_S(\Psi_T) + 6\sqrt{\frac{2/\delta}{2n}},$$

with probability at least $1 - \delta$. According to [3],

$$\mathbb{P}\big[ \{(x,y) : \exists \delta,\ \|\delta\|_\infty < \epsilon \text{ s.t. } \hat{y}(x+\delta) \neq y \} \big]$$
$$\leq \frac{2}{n} \sum_{i=1}^n \tilde{L}(f(x_i), y_i) + \frac{16}{n}$$
$$+ \frac{8}{n\gamma} \inf_{g \in \mathcal{G}} \sum_{i=1}^n \max_{\|\delta\|_\infty < \epsilon} \big| f(x_i + \delta) - g \circ h^T(x_i + \delta) \big|$$
$$+ 4\mathfrak{R}_S(\Psi_T) + 6\sqrt{\frac{2/\delta}{2n}}.$$

$\square$

## B.2 Proof of Theorem 4.2

In this section, we start with a binary classification task and logistic regression. Denote

$$f_\theta(x) = \theta^\top x, \quad g(f_\theta(x)) = \frac{1}{1 + \exp(-f_\theta(x))}, \quad h(f_\theta(x)) = \log\big(1 + \exp(f_\theta(x))\big).$$

Then the loss function $L$ can be rewritten as

$$L(f_\theta(x), y) = h(f_\theta(x)) - y f_\theta(x).$$

Notice that $\|\delta\|_\infty \leq \epsilon$ implies $\|\delta\|_2 \leq \epsilon\sqrt{d}$, and therefore

$$\max_{\|\delta\|_\infty \leq \epsilon} L(f_\theta(x), y) \leq \max_{\|\delta\|_2 \leq \epsilon\sqrt{d}} L(f_\theta(x), y),$$

where $d$ is the dimension of the input $x$. The standard empirical loss function can be written as

$$\mathcal{L}(f_\theta, \mathcal{D}) = \frac{1}{n} \sum_{i=1}^n L\big(f_\theta(\mathbf{x}_i), \mathbf{y}_i\big) = \frac{1}{n} \sum_{i=1}^n h\big(f_\theta(\mathbf{x}_i)\big) - \mathbf{y}_i f_\theta(\mathbf{x}_i),$$

where $\mathcal{D} = \{(\mathbf{x}_i, \mathbf{y}_i), i = 1, \ldots, n\}$. For a given $\epsilon > 0$, we consider the adversarial loss with $l_2$-attack of size $\epsilon\sqrt{d}$, that is,

$$\tilde{\mathcal{L}}(f_\theta, \mathcal{D}) = \frac{1}{n}\sum_{i=1}^{n} \max_{\|\delta_i\|_2 \le \epsilon\sqrt{d}} \tilde{L}\big(f_\theta(\mathbf{x}_i + \delta_i), \mathbf{y}_i\big) = \frac{1}{n}\sum_{i=1}^{n} \tilde{L}\big(f_\theta(\mathbf{x}_i), \mathbf{y}_i\big)$$

Consider the data-dependent parameter space:

$$\begin{aligned}\Theta \quad = \quad & \{\theta \in \mathbb{R}^d : \mathbf{y}_i f_\theta(\mathbf{x}_i) + (\mathbf{y}_i - 1)f_\theta(\mathbf{x}_i) \ge 0, \\ & \text{and} \quad |\mathbf{y}_i^* - g(f_\theta(\mathbf{x}_i))| \le \beta|\mathbf{y}_i - g(f_\theta(\mathbf{x}_i))|, \quad \text{for all } i = 1, \ldots, n\}.\end{aligned}$$

The first inequality considers the zero training error (0-1 loss). That is $f_\theta(\mathbf{x}) > 0$ when $\mathbf{y} = 1$ and $f_\theta(\mathbf{x}) \le 0$ if $\mathbf{y} = 0$. The second inequality constraints the distillation, i.e. $g(f_\theta(\mathbf{x}_i))$ is closed to the soft label given by the teacher model. Now we are ready to state the following theorem:

**Theorem.** *Suppose there exists a constant $c_x > 0$ such that $\|\mathbf{x}_i\|_2 > c_x\sqrt{d}$ for all $i \in \{1, \cdots, n\}$. Then, for any $\theta \in \Theta$, we have*

$$\tilde{\mathcal{L}}(f_\theta, \mathcal{D}) + \alpha\tilde{\mathcal{L}}(f_\theta, \mathcal{D}_{dis}) \le \mathcal{L}_{mix}(f_\theta, \mathcal{D}) + \alpha\mathcal{L}_{mix}(f_\theta, \mathcal{D}_{dis}),$$

*where the size of the adversarial attack $\epsilon$ is*

$$\epsilon = \frac{1 - \alpha\beta}{1 + \beta}c_x R\, \mathbb{E}_{\lambda \sim \tilde{P}_\lambda}[1 - \lambda], \quad \text{with} \quad R = \min_{i \in \{1, \ldots, n\}} |\cos(\theta, \mathbf{x}_i)|,$$

*and the distribution $\tilde{P}_\lambda$ is*

$$\tilde{P}_\lambda(\lambda) = \frac{\alpha}{\alpha + \beta}Beta(\alpha + 1, \beta) + \frac{\beta}{\alpha + \beta}Beta(\beta + 1, \alpha).$$

**Proof:** By the second order Taylor approximation,

$$L\big(f_\theta(x + \delta), y\big) \approx L\big(f_\theta(x), y\big) + \delta^\top \frac{\partial}{\partial x}L\big(f_\theta(x), y\big) + \frac{1}{2}\delta^\top \frac{\partial^2}{\partial x \partial x^\top}L\big(f_\theta(x), y\big)\delta.$$

Note that

$$\begin{aligned}\frac{\partial}{\partial x}L\big(f_\theta(x), y\big) \quad &= \quad \frac{\partial}{\partial x}\big(h(f_\theta(x)) - yf_\theta(x)\big) \\ &= \quad \frac{\partial}{\partial f}h(f_\theta(x))\frac{\partial}{\partial x}f_\theta(x) - y\frac{\partial}{\partial x}f_\theta(x) \\ &= \quad g\big(f_\theta(x)\big)\theta - y\theta \\ &= \quad \big(g(\theta^\top x) - y\big)\theta\end{aligned}$$

and

$$\begin{aligned}\frac{\partial^2}{\partial x \partial x^\top}L\big(f_\theta(x), y\big) \quad &= \quad \frac{\partial^2}{\partial x \partial x^\top}\big(h(f_\theta(x)) - yf_\theta(x)\big) \\ &= \quad \frac{\partial^2}{\partial f \partial f}h(f_\theta(x))\big(\frac{\partial}{\partial x}f_\theta(x)\big)^2 \\ &= \quad \frac{\partial}{\partial f}g(f_\theta(x))\theta\theta^\top \\ &= \quad g(\theta^\top x)\big(1 - g(\theta^\top x)\big)\theta\theta^\top.\end{aligned}$$

So we have

$$L\big(f_\theta(x + \delta), y\big) \approx L\big(f_\theta(x), y\big) + \big(g(\theta^\top x) - y\big)\theta^\top\delta + \frac{1}{2}g(\theta^\top x)\big(1 - g(\theta^\top x)\big)(\theta^\top\delta)^2$$

Furthermore

$$\begin{aligned}\tilde{\mathcal{L}}(f_\theta, \mathcal{D}) \quad \approx \quad & \frac{1}{n}\sum_{i=1}^{n}L\big(f_\theta(\mathbf{x}_i), \mathbf{y}_i\big) + \frac{1}{n}\sum_{i=1}^{n}\max_{\|\delta_i\|_2 \le \epsilon\sqrt{d}}\Big\{\big(g(\theta^\top\mathbf{x}_i) - \mathbf{y}_i\big)\theta^\top\delta_i \\ & + \frac{1}{2}g(\theta^\top\mathbf{x}_i)\big(1 - g(\theta^\top\mathbf{x}_i)\big)(\theta^\top\delta_i)^2\Big\} \\ =: \quad & \frac{1}{n}\sum_{i=1}^{n}L\big(f_\theta(\mathbf{x}_i), \mathbf{y}_i\big) + I_1.\end{aligned}$$

Similarly, for the data $\mathcal{D}_{dis} = \{(\mathbf{x}_i, \mathbf{y}_i^*), \mathbf{y}_i^* = f^T(\mathbf{x}_i), i = 1, \ldots, n\}$,

$$
\begin{aligned}
\tilde{\mathcal{L}}(f_\theta, \mathcal{D}_{dis}) \quad \approx \quad & \frac{1}{n} \sum_{i=1}^n L\big(f_\theta(\mathbf{x}_i), \mathbf{y}_i^*\big) + \frac{1}{n} \sum_{i=1}^n \max_{\|\delta_i\|_2 \le \epsilon\sqrt{d}} \Big\{ \big(g(\theta^\top \mathbf{x}_i) - \mathbf{y}_i^*\big)\theta^\top \delta_i \\
& + \frac{1}{2} g(\theta^\top \mathbf{x}_i)\big(1 - g(\theta^\top \mathbf{x}_i)\big)(\theta^\top \delta_i)^2 \Big\} \\
=: \quad & \frac{1}{n} \sum_{i=1}^n L\big(f_\theta(\mathbf{x}_i), \mathbf{y}_i^*\big) + I_2.
\end{aligned}
$$

Therefore,

$$
\tilde{\mathcal{L}}(f_\theta, \mathcal{D}) + \alpha \tilde{\mathcal{L}}(f_\theta, \mathcal{D}_{dis}) = \frac{1}{n} \sum_{i=1}^n L\big(f_\theta(\mathbf{x}_i), \mathbf{y}_i\big) + \frac{\alpha}{n} \sum_{i=1}^n L\big(f_\theta(\mathbf{x}_i), \mathbf{y}_i^*\big) + I_1 + \alpha I_2.
$$

Furthermore

$$
\begin{aligned}
I_1 + \alpha I_2 \quad \le \quad & \frac{1}{n} \sum_{i=1}^n \max_{\|\delta_i\|_2 \le \epsilon\sqrt{d}} \Big( \big(g(\theta^\top \mathbf{x}_i) - \mathbf{y}_i\big)\theta^\top \delta_i \Big) \\
& + \frac{\alpha}{n} \sum_{i=1}^n \max_{\|\delta_i\|_2 \le \epsilon\sqrt{d}} \Big( \big(g(\theta^\top \mathbf{x}_i) - \mathbf{y}_i^*\big)\theta^\top \delta_i \Big) \\
& + (1+\alpha)\frac{1}{2n} \sum_{i=1}^n \max_{\|\delta_i\|_2 \le \epsilon\sqrt{d}} g(\theta^\top \mathbf{x}_i)\big(1 - g(\theta^\top \mathbf{x}_i)\big)(\theta^\top \delta_i)^2 \\
= \quad & \frac{\epsilon\sqrt{d}}{n} \sum_{i=1}^n \big|g(\theta^\top \mathbf{x}_i) - \mathbf{y}_i\big|\|\theta\|_2 + \frac{\alpha\epsilon\sqrt{d}}{n} \sum_{i=1}^n \big|g(\theta^\top \mathbf{x}_i) - \mathbf{y}_i^*\big|\|\theta\|_2 \\
& + (1+\alpha)\epsilon^2 d\frac{1}{2n} \sum_{i=1}^n g(\theta^\top \mathbf{x}_i)\big(1 - g(\theta^\top \mathbf{x}_i)\big)\|\theta\|_2^2.
\end{aligned}
$$

Next we shall bound $\tilde{\mathcal{L}}(f_\theta, \mathcal{D}) + \alpha \tilde{\mathcal{L}}(f_\theta, \mathcal{D}_{dis})$ by mixup augmentation. Consider the following loss:

$$
\begin{aligned}
\mathcal{L}_{mix}(f_\theta, \mathcal{D}) \quad &= \quad \frac{1}{n^2} \sum_{i=1}^n \sum_{j=1}^n \mathbb{E}_{\lambda \sim P_\lambda} \big[ L(f_\theta(\mathbf{x}_{ij}(\lambda)), \mathbf{y}_{ij}(\lambda)) \big], \\
\mathcal{L}_{mix}(f_\theta, \mathcal{D}_{dis}) \quad &= \quad \frac{1}{n^2} \sum_{i=1}^n \sum_{j=1}^n \mathbb{E}_{\lambda \sim P_\lambda} \big[ L(f_\theta(\mathbf{x}_{ij}(\lambda)), \mathbf{y}_{ij}^*(\lambda)) \big],
\end{aligned}
$$

where $P_\lambda$ is a Beta distribution $\text{Beta}(\alpha, \beta)$, $\mathbf{y}_{ij}(\lambda) = \lambda\mathbf{y}_i + (1-\lambda)\mathbf{y}_j$ and $\mathbf{y}_{ij}^*(\lambda) = \lambda\mathbf{y}_i^* + (1-\lambda)\mathbf{y}_j^*$. We start with $\mathcal{L}_{mix}(f_\theta, \mathcal{D})$:

$$
\begin{aligned}
\mathcal{L}_{mix}(f_\theta, \mathcal{D}) \quad &= \quad \frac{1}{n^2} \sum_{i=1}^n \sum_{j=1}^n \mathbb{E}_{\lambda \sim P_\lambda} \big[ h(\theta^\top \mathbf{x}_{ij}(\lambda)) - \mathbf{y}_{ij}(\lambda)\theta^\top \mathbf{x}_{ij}(\lambda) \big] \\
&= \quad \frac{1}{n^2} \sum_{i=1}^n \sum_{j=1}^n \mathbb{E}_{\lambda \sim P_\lambda} \Big\{ \lambda\big[ h(\theta^\top \mathbf{x}_{ij}(\lambda)) - \mathbf{y}_i\theta^\top \mathbf{x}_{ij}(\lambda) \big] \\
&\qquad + (1-\lambda)\big[ h(\theta^\top \mathbf{x}_{ij}(\lambda)) - \mathbf{y}_j\theta^\top \mathbf{x}_{ij}(\lambda) \big] \Big\}
\end{aligned}
$$

We introduce a 0-1 random variable $B$ such that the conditional distribution of $B$ given $\lambda$ is

$$
P(B = 1|\lambda) = \lambda, \quad \text{and} \quad P(B = 0|\lambda) = 1 - \lambda.
$$

Rewrite $\mathcal{L}_{mix}(f_\theta, \mathcal{D})$ as

$$
\begin{aligned}
\mathcal{L}_{mix}(f_\theta, \mathcal{D}) \quad &= \quad \frac{1}{n^2} \sum_{i=1}^n \sum_{j=1}^n \mathbb{E}_{\lambda \sim P_\lambda} \Big\{ \mathbb{E}_{B|\lambda} \Big\{ B\big[ h(\theta^\top \mathbf{x}_{ij}(\lambda)) - \mathbf{y}_i\theta^\top \mathbf{x}_{ij}(\lambda) \big] \\
&\qquad + (1-B)\big[ h(\theta^\top \mathbf{x}_{ij}(\lambda)) - \mathbf{y}_j\theta^\top \mathbf{x}_{ij}(\lambda) \big] \Big\} \Big\}.
\end{aligned}
$$

Notice that

$$P(\lambda, B = 1) = P(B = 1|\lambda)P(\lambda) = \frac{\lambda^\alpha(1-\lambda)^{\beta-1}}{B(\alpha,\beta)} = \frac{\lambda^\alpha(1-\lambda)^{\beta-1}}{B(\alpha+1,\beta)} \times \frac{\alpha}{\alpha+\beta}$$

$$P(\lambda, B = 0) = P(B = 0|\lambda)P(\lambda) = \frac{\lambda^{\alpha-1}(1-\lambda)^{\beta}}{B(\alpha,\beta)} = \frac{\lambda^{\alpha-1}(1-\lambda)^{\beta}}{B(\alpha,\beta+1)} \times \frac{\beta}{\alpha+\beta}$$

where $B(\cdot,\cdot)$ is the Beta function. Thus the marginal distribution of $B$ is

$$P(B = 1) = \frac{\alpha}{\alpha+\beta} \quad \text{and} \quad P(B = 0) = \frac{\beta}{\alpha+\beta}$$

and the conditional distribution of $\lambda$ given $B$ is

$$P(\lambda|B = 1) = \text{Beta}(\alpha+1,\beta) \quad \text{and} \quad P(\lambda|B = 0) = \text{Beta}(\alpha,\beta+1).$$

Then we have

$$
\begin{aligned}
\mathcal{L}_{mix}(f_\theta, \mathcal{D}) &= \frac{1}{n^2}\sum_{i=1}^{n}\sum_{j=1}^{n}\mathbb{E}_B\Big\{\mathbb{E}_{\lambda|B}\Big\{B\big[h(\theta^\top\mathbf{x}_{ij}(\lambda)) - \mathbf{y}_i\theta^\top\mathbf{x}_{ij}(\lambda)\big] \\
&\quad + (1-B)\big[h(\theta^\top\mathbf{x}_{ij}(\lambda)) - \mathbf{y}_j\theta^\top\mathbf{x}_{ij}(\lambda)\big]\Big\}\Big\} \\
&= \frac{1}{n^2}\sum_{i=1}^{n}\sum_{j=1}^{n}\Big\{\frac{\alpha}{\alpha+\beta}\mathbb{E}_{\lambda\sim\text{Beta}(\alpha+1,\beta)}\big[h(\theta^\top\mathbf{x}_{ij}(\lambda)) - \mathbf{y}_i\theta^\top\mathbf{x}_{ij}(\lambda)\big] \\
&\quad + \frac{\beta}{\alpha+\beta}\mathbb{E}_{\lambda\sim\text{Beta}(\alpha,\beta+1)}\big[h(\theta^\top\mathbf{x}_{ij}(\lambda)) - \mathbf{y}_j\theta^\top\mathbf{x}_{ij}(\lambda)\big]\Big\}.
\end{aligned}
$$

In addition, let

$$\lambda' = 1 - \lambda \sim \text{Beta}(\beta+1,\alpha).$$

So we can transform the random variable from $\lambda$ to $\lambda'$ and have

$$
\begin{aligned}
&\mathbb{E}_{\lambda\sim\text{Beta}(\alpha,\beta+1)}\big[h(\theta^\top\mathbf{x}_{ij}(\lambda)) - \mathbf{y}_j\theta^\top\mathbf{x}_{ij}(\lambda)\big] \\
&= \mathbb{E}_{\lambda'\sim\text{Beta}(\beta+1,\alpha)}\big[h(\theta^\top\mathbf{x}_{ji}(\lambda')) - \mathbf{y}_j\theta^\top\mathbf{x}_{ji}(\lambda')\big].
\end{aligned}
$$

Plug this equation into $\mathcal{L}_{mix}(f_\theta, \mathcal{D})$,

$$
\begin{aligned}
\mathcal{L}_{mix}(f_\theta, \mathcal{D}) &= \frac{1}{n^2}\sum_{i=1}^{n}\sum_{j=1}^{n}\Big\{\frac{\alpha}{\alpha+\beta}\mathbb{E}_{\lambda\sim\text{Beta}(\alpha+1,\beta)}\big[h(\theta^\top\mathbf{x}_{ij}(\lambda)) - \mathbf{y}_i\theta^\top\mathbf{x}_{ij}(\lambda)\big] \\
&\quad + \frac{\beta}{\alpha+\beta}\mathbb{E}_{\lambda\sim\text{Beta}(\beta+1,\alpha)}\big[h(\theta^\top\mathbf{x}_{ij}(\lambda)) - \mathbf{y}_i\theta^\top\mathbf{x}_{ij}(\lambda)\big]\Big\} \\
&= \frac{1}{n^2}\sum_{i=1}^{n}\sum_{j=1}^{n}\mathbb{E}_{\lambda\sim\tilde{P}_\lambda}\big[h(\theta^\top\mathbf{x}_{ij}(\lambda)) - \mathbf{y}_i\theta^\top\mathbf{x}_{ij}(\lambda)\big],
\end{aligned}
$$

where

$$\tilde{P}_\lambda(\lambda) = \frac{\alpha}{\alpha+\beta}\text{Beta}(\alpha+1,\beta) + \frac{\beta}{\alpha+\beta}\text{Beta}(\beta+1,\alpha).$$

To proceed further, we denote $P_n(x)$ as the empirical distribution of $\mathbf{x}$ induced by the training sample. Then $I_2$ can be rewritten as

$$
\begin{aligned}
\mathcal{L}_{mix}(f_\theta, \mathcal{D}) &= \frac{1}{n}\sum_{i=1}^{n}\mathbb{E}_{\mathbf{x}\sim P_n}\mathbb{E}_{\lambda\sim\tilde{P}_\lambda}\Big[h(\theta^\top(\lambda\mathbf{x}_i + (1-\lambda)\mathbf{x})) \\
&\quad - \mathbf{y}_i\theta^\top(\lambda\mathbf{x}_i + (1-\lambda)\mathbf{x})\Big] \\
&= \frac{1}{n}\sum_{i=1}^{n}\mathbb{E}_{\mathbf{x}\sim P_n}\mathbb{E}_{\lambda\sim\tilde{P}_\lambda}\psi_i(\gamma),
\end{aligned}
$$

where $\gamma = 1 - \lambda$ and

$$\psi_i(\gamma) = h(\theta^\top((1-\gamma)\mathbf{x}_i + \gamma\mathbf{x})) - \mathbf{y}_i\theta^\top((1-\gamma)\mathbf{x}_i + \gamma\mathbf{x}).$$

By the second order Taylor expansion,

$$\psi_i(\gamma) = \psi_i(0) + \psi_i'(0)\gamma + \frac{1}{2}\psi_i''(0)\gamma^2 + O(\gamma^3).$$

Furthermore,

$$
\begin{aligned}
\psi_i'(0) &= h'(\theta^\top\mathbf{x}_i)\theta^\top(\mathbf{x} - \mathbf{x}_i) - \mathbf{y}_i\theta^\top(\mathbf{x} - \mathbf{x}_i) \\
&= \big(g(\theta^\mathrm{T}\mathbf{x}_i) - \mathbf{y}_i\big)\theta^\top(\mathbf{x} - \mathbf{x}_i),
\end{aligned}
$$

and

$$
\begin{aligned}
\psi_i''(0) &= h''(\theta^\top\mathbf{x}_i)[\theta^\top(\mathbf{x} - \mathbf{x}_i)]^2 \\
&= g(\theta^\mathrm{T}\mathbf{x}_i)\big(1 - g(\theta^\mathrm{T}\mathbf{x}_i)\big)[\theta^\top(\mathbf{x} - \mathbf{x}_i)]^2.
\end{aligned}
$$

Thus we have $\mathcal{L}_{mix}(f_\theta, \mathcal{D}) = \frac{1}{n}\sum_{i=1}^n L(f_\theta(\mathbf{x}_i), \mathbf{y}_i) + I_3 + I_4$, where

$$
\begin{aligned}
I_3 &= \mathbb{E}_{\lambda \sim \tilde{P}_\lambda}[1-\lambda]\frac{1}{n}\sum_{i=1}^n \mathbb{E}_{\mathbf{x}\sim P_n}\big[\big(g(\theta^\mathrm{T}\mathbf{x}_i) - \mathbf{y}_i\big)\theta^\top(\mathbf{x} - \mathbf{x}_i)\big], \\
I_4 &= \mathbb{E}_{\lambda \sim \tilde{P}_\lambda}[(1-\lambda)^2]\frac{1}{2n}\sum_{i=1}^n \mathbb{E}_{\mathbf{x}\sim P_n}\big[g(\theta^\mathrm{T}\mathbf{x}_i)\big(1 - g(\theta^\mathrm{T}\mathbf{x}_i)\big)[\theta^\top(\mathbf{x} - \mathbf{x}_i)]^2\big].
\end{aligned}
$$

Similarly, for the data $\mathcal{D}_{dis} = \{(\mathbf{x}_i, \mathbf{y}_i^*), \mathbf{y}_i^* = f^T(\mathbf{x}_i), i = 1, \ldots, n\}$, the following decomposition also holds: $\mathcal{L}_{mix}(f_\theta, \mathcal{D}_{dis}) = \frac{1}{n}\sum_{i=1}^n L(f_\theta(\mathbf{x}_i), \mathbf{y}_i^*) + I_5 + I_4$, where

$$I_5 = \mathbb{E}_{\lambda \sim \tilde{P}_\lambda}[1-\lambda]\frac{1}{n}\sum_{i=1}^n \mathbb{E}_{\mathbf{x}\sim P_n}\big[\big(g(\theta^\mathrm{T}\mathbf{x}_i) - \mathbf{y}_i^*\big)\theta^\top(\mathbf{x} - \mathbf{x}_i)\big].$$

Therefore,

$$
\begin{aligned}
\mathcal{L}_{mix}(f_\theta, \mathcal{D}) + \alpha\mathcal{L}_{mix}(f_\theta, \mathcal{D}_{dis}) &= \frac{1}{n}\sum_{i=1}^n L(f_\theta(\mathbf{x}_i), \mathbf{y}_i) + \frac{\alpha}{n}\sum_{i=1}^n L(f_\theta(\mathbf{x}_i), \mathbf{y}_i^*) \\
&\quad + (I_3 + \alpha I_5) + (1+\alpha)I_4.
\end{aligned}
$$

For the term $I_3 + \alpha I_5$, we have

$$
\begin{aligned}
I_3 + \alpha I_5 &= \mathbb{E}_{\lambda \sim \tilde{P}_\lambda}[1-\lambda]\frac{1}{n}\sum_{i=1}^n \big((1+\alpha)g(\theta^\mathrm{T}\mathbf{x}_i) - \mathbf{y}_i - \alpha\mathbf{y}_i^*\big)\theta^\top\big(\mathbb{E}_{\mathbf{x}\sim P_n}(\mathbf{x}) - \mathbf{x}_i\big) \\
&= \mathbb{E}_{\lambda \sim \tilde{P}_\lambda}[1-\lambda]\frac{1}{n}\sum_{i=1}^n \big(\mathbf{y}_i + \alpha\mathbf{y}_i^* - (1+\alpha)g(\theta^\mathrm{T}\mathbf{x}_i)\big)\theta^\top\mathbf{x}_i \\
&= \mathbb{E}_{\lambda \sim \tilde{P}_\lambda}[1-\lambda]\frac{1}{n}\sum_{i=1}^n \big|\mathbf{y}_i + \alpha\mathbf{y}_i^* - (1+\alpha)g(\theta^\mathrm{T}\mathbf{x}_i)\big|\|\theta\|_2\|\mathbf{x}_i\|_2|\cos(\theta, \mathbf{x}_i)| \\
&= \mathbb{E}_{\lambda \sim \tilde{P}_\lambda}[1-\lambda]\frac{1}{n}\sum_{i=1}^n \big||\mathbf{y}_i - g(\theta^\mathrm{T}\mathbf{x}_i)| - \alpha|\mathbf{y}_i^* - g(\theta^\mathrm{T}\mathbf{x}_i)|\big|\|\theta\|_2\|\mathbf{x}_i\|_2|\cos(\theta, \mathbf{x}_i)| \\
&\geq R_i c_x\sqrt{d}\,\mathbb{E}_{\lambda \sim \tilde{P}_\lambda}[1-\lambda]\frac{1}{n}\sum_{i=1}^n (1-\alpha k)\big|\mathbf{y}_i - g(\theta^\mathrm{T}\mathbf{x}_i)\big|\|\theta\|_2 \\
&\geq \epsilon\sqrt{d}\frac{1}{n}\sum_{i=1}^n (1+k)\big|\mathbf{y}_i - g(\theta^\mathrm{T}\mathbf{x}_i)\big|\|\theta\|_2 \\
&\geq \frac{1}{n}\sum_{i=1}^n \max_{\|\delta_i\|_2 \leq \epsilon\sqrt{d}} \big(\mathbf{y}_i - g(\theta^\mathrm{T}\mathbf{x}_i)\big)\theta^\top\delta_i + \frac{\alpha}{n}\sum_{i=1}^n \max_{\|\delta_i\|_2 \leq \epsilon\sqrt{d}} \big(\mathbf{y}_i^* - g(\theta^\mathrm{T}\mathbf{x}_i)\big)\theta^\top\delta_i.
\end{aligned}
$$

Now we turn to $I_4$:

$$
\begin{aligned}
I_4 &= \mathbb{E}_{\lambda \sim \tilde{P}_\lambda}[(1-\lambda)^2]\frac{1}{2n}\sum_{i=1}^{n} g(\theta^{\mathrm{T}}\mathbf{x}_i)\big(1-g(\theta^{\mathrm{T}}\mathbf{x}_i)\big)\theta^\top \mathbb{E}_{\mathbf{x}\sim P_n}[(\mathbf{x}-\mathbf{x}_i)(\mathbf{x}-\mathbf{x}_i)^\top]\theta \\
&= \mathbb{E}_{\lambda \sim \tilde{P}_\lambda}[(1-\lambda)^2]\frac{1}{2n}\sum_{i=1}^{n} g(\theta^{\mathrm{T}}\mathbf{x}_i)\big(1-g(\theta^{\mathrm{T}}\mathbf{x}_i)\big)\theta^\top \big(\mathbb{E}_{\mathbf{x}\sim P_n}(\mathbf{x}\mathbf{x}^\top)+\mathbf{x}_i\mathbf{x}_i^\top\big)\theta \\
&\geq \mathbb{E}_{\lambda \sim \tilde{P}_\lambda}[(1-\lambda)^2]\frac{1}{2n}\sum_{i=1}^{n} g(\theta^{\mathrm{T}}\mathbf{x}_i)\big(1-g(\theta^{\mathrm{T}}\mathbf{x}_i)\big)\theta^\top \mathbf{x}_i\mathbf{x}_i^\top \theta \\
&= R_i^2 c_x^2 d\; \mathbb{E}_{\lambda \sim \tilde{P}_\lambda}[(1-\lambda)^2]\frac{1}{2n}\sum_{i=1}^{n} g(\theta^{\mathrm{T}}\mathbf{x}_i)\big(1-g(\theta^{\mathrm{T}}\mathbf{x}_i)\big)\|\theta\|_2^2 \\
&\geq \epsilon^2 d\frac{1}{2n}\sum_{i=1}^{n} g(\theta^{\mathrm{T}}\mathbf{x}_i)\big(1-g(\theta^{\mathrm{T}}\mathbf{x}_i)\big)\|\theta\|_2^2 \\
&= \frac{1}{n}\sum_{i=1}^{n}\max_{\|\delta_i\|_2\leq \epsilon\sqrt{d}} g(\theta^\top\mathbf{x}_i)\big(1-g(\theta^\top\mathbf{x}_i)\big)(\theta^\top\delta_i)^2.
\end{aligned}
$$

Combining the results of $I_3+\alpha I_5$ and $I_4$, we know that $I_3+\alpha I_5 \geq I_1+\alpha I_2$. Furthermore, we have

$$
\tilde{\mathcal{L}}(f_\theta,\mathcal{D}) + \alpha\tilde{\mathcal{L}}(f_\theta,\mathcal{D}_{dis}) \leq \mathcal{L}_{mix}(f_\theta,\mathcal{D}) + \alpha\mathcal{L}_{mix}(f_\theta,\mathcal{D}_{dis}).
$$

$\square$

Next we extend the above theorem to the case of neural networks with ReLU activation and max-pooling. We still consider the binary classification task with the logistic loss and take $f_\theta(x)$ to be a fully connected neural network with ReLU activation function or max-pooling:

$$
f_\theta(x) = \mathbf{a}^\top \sigma\big(W_{N-1}\cdots \sigma(W_2\sigma(W_1 x))\big),
$$

where $\sigma(\cdot)$ is a nonlinear function that consists of ReLU activation and max pooling, each $W_i$ is a matrix, and $\mathbf{a}$ is a column vector: i.e., $\theta$ consists of $\{W_i, i=1,\ldots,W-1\}$ and $\mathbf{a}$. According to the derivatives of ReLU and max-pooling, the function $f_\theta$ satisfies that $\nabla_x^2 f_\theta(x) = 0$ and $f_\theta(x) = \nabla_x f_\theta(x)^\top x$ almost everywhere. Therefore a fully connected neural networks with ReLU activation functions and max-pooling can be locally approximated by a linear function. By the results of logistic regression, we have the following theorem:

**Theorem**. *Suppose $f_\theta(x) = \nabla_x f_\theta(x)^\top x$, $\nabla_x^2 f_\theta(x) = 0$ and there exists a constant $c_x > 0$ such that $\|\mathbf{x}_i\|_2 > c_x\sqrt{d}$ for all $i \in \{1,\cdots,n\}$. Then, for any $\theta \in \Theta$, we have*

$$
\tilde{\mathcal{L}}(f_\theta,\mathcal{D}) + \alpha\tilde{\mathcal{L}}(f_\theta,\mathcal{D}_{dis}) \leq \mathcal{L}_{mix}(f_\theta,\mathcal{D}) + \alpha\mathcal{L}_{mix}(f_\theta,\mathcal{D}_{dis}),
$$

*where the size of the adversarial attack $\epsilon$ is*

$$
\epsilon = \frac{1-\alpha\beta}{1+\beta}c_x R\, \mathbb{E}_{\lambda\sim\tilde{P}_\lambda}[1-\lambda], \quad with \quad R = \min_{i\in\{1,\ldots,n\}}|\cos(\nabla_x f_\theta(x),\mathbf{x}_i)|,
$$

*and the distribution $\tilde{P}_\lambda$ is*

$$
\tilde{P}_\lambda(\lambda) = \frac{\alpha}{\alpha+\beta}Beta(\alpha+1,\beta) + \frac{\beta}{\alpha+\beta}Beta(\beta+1,\alpha).
$$

$\square$