# OpenReview forum: "MixACM: Mixup-Based Robustness Transfer via Distillation of Activated Channel Maps"
_NeurIPS.cc/2021/Conference — NeurIPS 2021 Poster_

### Official Review · Reviewer_up6G · 2021-07-11

**Rating:** 7
**Confidence:** 4

**Summary:**

This paper aims to transfer the robustness from a robust teacher model to student model with the help of mixup augmentation. It proposes the MixACM method to transfer the robustness from teacher model to student model by matching their channel-wise activation maps.  From the theoretical perspective, it showed that adversarial loss can be bounded with the natural loss and distance between the student and teacher model on mixup examples. Empirically, extensive experiments are conducted including robustness transfer, distillation and transfer learning to demonstrate the effectiveness of the method.

**Limitations And Societal Impact:**

The paper have discussed the limitation in the supplemental materials, the robustness transfer deteriorates significantly if depth or width of student model is reduced significantly.

**Main Review:**

Strengths:

This paper is well-written and conducts solid experiments to demonstrate the effectiveness. From Table 1, we can see that the robustness of the student model outperforms the teacher models, and is also comparable to most state-of-the-art adversarial training methods. It also compares with the latest robustness distillation methods, IGAM and RKD, it seems that it sets a new benchmark in this robustness distillation. I am glad to see that the authors evaluate the robustness under AutoAttack and also give the distillation results on ImageNet.

Suggestions:

From Table 6 and Figure 7 in supplemental materials, it seems that the performance gained in student models mainly comes from the feature distillation. I think it would be better to give the experiments of ACM on different layers. Does the low level or high level features matter?


**Time Spent Reviewing:**

6 hours

---

> ### Author Response · Authors · 2021-08-10
> **Reply to R4**
>
> Thank you for the positive feedback and interesting suggestion. We are happy that you find our paper well-written and having solid experiments. We have conducted an ablation study in light of your recommendation.
>
> **Q1:** From Table 6 and Figure 7 in supplemental materials, it seems that the performance gained in student models mainly comes from the feature distillation. I think it would be better to give the experiments of ACM on different layers. Does the low level or high level features matter?
>
> **A1:** To understand the role of low, mid, and high-level features, we performed experiments on CIFAR10 by progressively changing blocks used for distillation. For this ablation study, we kept all the standard settings reported in the paper (supplementary material section A.1). Our correspondence of blocks and features is as follows: block 2: low-level features; block 3: mid-level features; block 4: high-level features. Please note that block 1 corresponds to the output of the first layer only. Therefore, we do not call it low-level features. The results are shown in the following table.
>
> | Features Used          | Accuracy | Robustness |
> | ---------------------- | -------- | ---------- |
> | Low-Level (2)          | 86.36    | 17.24      |
> | Mid-Level (3)          | 88.30    | 39.37      |
> | High-Level (4)         | 91.18    | 33.13      |
> |                        |          |            |
> | Low + Mid Level (2+3)  | 88.15    | 41.92      |
> | Mid + High Level (3+4) | 90.69    | 52.79      |
> |                        |          |            |
> | First Layer Only (1)   | 86.00    | 0.31       |
> | No First Layer (2+3+4) | 90.69    | 56.15      |
> |                        |          |            |
> | All Features           | 90.76    | 56.65      |
>
>
> As shown in the table, mid-level features play a more important role in robustness transfer and high-level features play a crucial role in accuracy transfer. The robustness of the student is 39.37% when we only use mid-level features, but it decreases to 33.13% when high-level features are utilized alone. On the other hand, clean accuracy improves when we only use high-level features: 88.30% with mid-level compared to 91.18% with high-level features. In addition, a combination of mid and high-level features is enough to get close to optimal robustness and accuracy. But the addition of low-level features improves robustness even further.
>
> Apart from these low, mid, and high-level features, we also used the output of the first layer in the proposed loss function. Our experiments above show that the improvement brought by first layer distillation is relatively small. Specifically, the addition of the first layer in the above-mentioned experiments brings $\leq 1$​​% improvement for robustness.
>
> In summary, all level features (low, mid, high level) improve robustness and accuracy. However, mid-level features seem to be more critical for robustness and high-level features for accuracy.

---

> > ### Comment · Reviewer_up6G · 2021-09-02
> > **Thanks for your response**
> >
> > Thank you very much for your response. I suggest that the authors can add this discussion in the final version. I leave my score unchanged.

---

### Official Review · Reviewer_8Aug · 2021-07-16

**Rating:** 6
**Confidence:** 4

**Summary:**

This paper proposes a method for robustness transfer. Based on the theoretical analysis, an activated
channel maps-based distillation along with mixup is proposed to replace the traditional adversarial loss for
robustness transfer. Experiments on CIFAR10, CIFAR100, and ImageNet datasets with different settings
demonstrate the proposed method achieves outstanding performance.

**Limitations And Societal Impact:**

Please refer to the Main Review part for detail.
I encourage the authors to clarify the novelty more clearly in the rebuttal.

**Main Review:**

1. The writing is relatively good and the paper is easy to understand.
2. The motivation is very interesting, and the theoretical analysis well explains the principle of proposed
methods.
3. Some writing mistakes should be fixed, for example, in Line 35, ‘compare’ should be ‘compared’.
4. The proposed method seems to be a simple combination of some existing methods such as
knowledge distillation, mixup and activated channel maps. Considering that the proposed method is
essentially a distillation method, compared to previous works for distillation, this paper may lack
novelty.
5. The experimental part lacks some essential ablation studies. For example, what’s the performance of
directly distillating the middle features? The performance gain gotten from the activated channels
maps should be presented if the teacher and student use the same dataset.

**Time Spent Reviewing:**

4

---

> ### Author Response · Authors · 2021-08-10
> **Reply to R3**
>
> We thank you for your time and insightful questions. We are encouraged by positive feedback.
>
> ____
>
> **Q1:** The motivation is very interesting, and the theoretical analysis well explains the principle of proposed methods. The proposed method seems to be a simple combination of some existing methods such as knowledge distillation, mixup and activated channel maps. Considering that the proposed method is essentially a distillation method, compared to previous works for distillation, this paper may lack novelty. Limitations: I encourage the authors to clarify the novelty more clearly in the rebuttal.
>
> **A1:** Knowledge distillation (KD) is a broad and well-explored area of research consisting of a large body of work. The KD works aim to transfer knowledge from a teacher to a student to enhance or improve *clean accuracy* under various circumstances, e.g., compact students, different datasets, unlabeled datasets, etc. Robustness distillation, however, is a relatively new direction that is about transferring robustness from a teacher to a student, ideally without producing adversarial examples [1, 2]. The existing robustness transfer methods either match gradients (w.r.t. input) of teacher and student model along with a discriminator [1], or they match logits produced by teachers on adversarial examples [2].
>
> Our method, on the other hand, proposes to use intermediate features produced on mixup augmented examples. With the activated channel maps, it can not only achieve better robustness, and clean accuracy compared to existing methods, but it does so without using adversarial examples, any extra gradient steps, fine-tuning or discriminator-based learning. Furthermore, our method can do distillation with different sizes of intermediate features making it possible to significantly shrink the size of the student. Also, the robustness of ACM is comparable to state-of-the-art adversarial training methods while also improving clean accuracy.
>
> For robustness distillation from intermediate features, we have proposed to use activated channel maps of teacher's intermediate features to distill robustness. While the idea of the activated channel maps is motivated based on the observations in [3], it is substantially different from it. Specifically, ACM loss transfers the pattern of activated channels from a teacher to a student. On the other hand, [3] suppresses channels that do not contribute to the ground-truth output in adversarial training. The idea of finding the activated channels is also significantly different from the way [3] finds the contribution of each channel. Specifically, we extract activated channel maps by applying a simple and computationally cheap function without using label information. In [3], a fully connected (FC) layer is applied for each activation layer. The ground-truth class-related weights of this FC layer determine the importance.
>
> Finally, recent works do show a link between robustness and mixup [4, 5]. However, our work does it for robustness distillation. Specifically, we first theoretically showed the importance of mixup in robustness transfer and then demonstrated it experimentally.
>
> ____
>
> **Q2:** The experimental part lacks some essential ablation studies. For example, what’s the performance of directly distillating the middle features? The performance gain gotten from the activated channels maps should be presented if the teacher and student use the same dataset.
>
> **A2:** The purpose of ACM loss is to match activated channel maps of teacher and student. It is possible to distill robustness by directly matching intermediate features of teacher and student. However, this direct way of distillation overlooks differences between the teacher and the student such as structure, number of channels, size of activations, how and on what data teacher is trained, etc.
>
> To answer the question, we also have conducted an ablation study comparing direct distillation with ACM-based distillation while progressively increasing differences between the teacher and the student. We have kept all the standard settings shared in the paper and used similar settings for direct distillation for a fair comparison. The following table reports the results. When teacher and student are similar, ACM performs slightly better than direct distillation (56.65% vs. 56.12%). However, when the number of channels of teacher and student is different, the performance gap increases (48.75% vs. 44.95%). This gap increases further when both channels and the number of layers are different (47.18% vs. 41.90%). A similar gap is also visible in terms of clean accuracy for all these cases.
>
> To further explore the effect of this difference, we also performed one experiment under transfer learning settings for CIFAR100 (Table 5 in the paper). Here, the teacher is trained on a different dataset (ImageNet), so the difference between the two models is larger. The performance gap is also wider. ACM outperforms direct distillation significantly (clean accuracy: 65.69% vs. 57.86% and robustness: 24.14% vs. 16.20%).
>
> | Method                | Teacher  | Student  | Accuracy  | Robustness |
> | --------------------- | -------- | -------- | --------- | ---------- |
> |**Distillation for CIFAR10**|||||
> | Direct                | WRN28-10 | WRN34-10 | 89.98     | 56.12      |
> | ACM                   | WRN28-10 | WRN34-10 | **90.76** | **56.65**  |
> |                       |          |          |           |            |
> | Direct                | WRN28-10 | WRN28-5  | 88.46     | 44.95      |
> | ACM                   | WRN28-10 | WRN28-5  | **90.26** | **48.75**  |
> |                       |          |          |           |            |
> | Direct                | WRN34-20 | WRN16-10 | 84.27     | 41.90      |
> | ACM                   | WRN34-20 | WRN16-10 | **86.31** | **47.18**  |
> |**Transfer Learning for CIFAR100**|||||
> | Direct                | WRN28-10 | WRN34-10 | 57.86     | 16.20      |
> | ACM                   | WRN28-10 | WRN34-10 | **65.69** | **24.14**  |
>
>
>
> ___
>
> **Q3**: Some writing mistakes should be fixed, for example, in Line 35, ‘compare’ should be ‘compared’.
>
> **A3**: Thank you for pointing this out. We have modified the manuscript to rectify these issues.
>
> ____
>
> **References**
>
> [1] Chan et al., "What it Thinks is Important is Important: Robustness Transfers through Input Gradients", Oral at CVPR'20.
>
> [2] Goldblum et al., "Adversarially Robust Distillation", AAAI'20
>
> [3] Bai et al., "Improving Adversarial Robustness via Channel-wise Activation Suppressing", Spotlight at ICLR'21
>
> [4] Linjun et al., "How Does Mixup Help With Robustness and Generalization?", ICLR'21
>
> [5] Zhang et al., "mixup: Beyond Empirical Risk Minimization", ICLR'18

---

> > ### Comment · Reviewer_8Aug · 2021-08-30
> > **Thanks for the response!**
> >
> > The explanation addressed most of my concerns. I would like to increase my score by 1 point (from 5 to 6). However, the writing could be further improved.

---

> > > ### Author Response · Authors · 2021-08-30
> > > **Thank You for Increasing the Score**
> > >
> > > Thank you very much for increasing the score. We are glad that our explanation addressed your concerns. We are improving the manuscript according to the feedback given in the reviews.

---

### Official Review · Reviewer_d7Ao · 2021-07-17

**Rating:** 6
**Confidence:** 3

**Summary:**

The submission proposes a novel robustness transfer method, named MixACM, to transfer robustness from a robust teacher to a student by matching activated channel maps generated without expensive adversarial perturbations. Extensive experiments in cifar and imagenet show the improvements.

**Main Review:**

1. The paper is well written and easy to follow.
2. in Line 171, theorem 1 starts with the assumption that a fully connected neural network with ReLU or max-pooling. However, in Fig 1, the proposed method has the blocks of resnet. It is not the same with the assumption in theorem.
3. Fig 1 also shows the intermediate features. But this also didn't appear in theorem 1. The author can give more explanations on this part.

**Time Spent Reviewing:**

3 hours

---

> ### Author Response · Authors · 2021-08-10
> **Reply to R2**
>
> We thank you for your time and valuable feedback. We answer the questions here.
>
> **Q1:** In Line 171, theorem 1 starts with the assumption that a fully connected neural network with ReLU or max-pooling. However, in Fig. 1, the proposed method has the blocks of resnet. It is not the same with the assumption in theorem. Fig 1 also shows the intermediate features. But this also didn't appear in theorem 1. The author can give more explanations on this part.
>
> **A1:** Theorem 4.1 considers a general hypothetical space and holds for both fully connected neural networks and ResNet. So this is a general result. For Theorem 4.2, we assume the model $f$ is a fully connected neural network with ReLU or max-pooling. In the proof of Theorem 4.2 (Appendix B.2.), we start with logistic regression and then use the first-order expansion to approximate a fully connected neural network. Therefore, Theorem 4.2 is derived from a basic result under simple linear models. Intuitively, we think this theorem can be extended to more complex neural networks, although we do not have a formal guarantee. The heuristic understanding is as follows. We can use Neural Tangent Kernel [1] to approximate the inference procedure of wide neural networks with the inference procedure of kernel methods. Furthermore, the NTK can be studied for various architectures, e.g., CNN, RNN, and Transformer [2, 3, 4]. Hence, we believe that the preliminary result in Theorem 4.2 is a good start point to inspire a more complex distillation method.
>
> We would also like to point out that the theoretical analysis is often performed on relatively simple models. This indirect way of theoretical study is prevalent in the community. We give two examples from robustness literature here. In [5], the authors studied a simple Gaussian model for theoretical analysis, and these insights are extended to deep models. Similarly, [6] showed adversarial Rademacher complexity for binary and multi-class binary classifiers and one-hidden layer ReLU network. These theoretical results are validated by experimental results on more complex deep models.
>
>
>
> **References**:
>
> [1] Arthur et al. "Neural Tangent Kernel: Convergence and Generalization in Neural Networks", NeurIPS'18.
>
> [2] Sanjeev et al. "On exact computation with an infinitely wide neural net", NeurIPS'19.
>
> [3] Yang, Greg., "Scaling limits of wide neural networks with weight sharing: Gaussian process behavior, gradient independence, and neural tangent kernel derivation", Arxiv'19.
>
> [4] Hron et al. "Infinite attention: NNGP and NTK for deep attention networks", ICML'20.
>
> [5] Carmon et al., "Unlabeled Data Improves Adversarial Robustness", NeurIPS'19
>
> [6] Yin et al., "Rademacher Complexity for Adversarially Robust Generalization", ICML'19

---

> ### Comment · Reviewer_d7Ao · 2021-09-01
> **thanks for the responses**
>
> Thanks for the responses. Concerns can be solved by improving the writing parts. I will keep the score unchanged as 6.

---

### Official Review · Reviewer_Fo9x · 2021-07-18

**Rating:** 6
**Confidence:** 4

**Summary:**

This paper proposes a robustness transfer method that combines mixup data augmentation, distillation from both prediction and intermediate features.

It is motivated by the theoretical insight that robust test error can be decomposed into adversarial loss plus distillation loss, and both of them can be upper bounded by mixup loss. Motivated by some other previous work, the KL divergence with the teacher model’s prediction and L2 distance with the teacher model’s intermediate activated feature maps are also added to the final loss of the student model. So the loss is composed of three terms: mixup loss, KL divergence between predictions, and L2 distance between the intermediate activation maps.

Experiments on popular datasets such as CIFAR-10, CIFAR-100 and ImageNet demonstrate it’s superiority compared with a variety of baselines.


**Limitations And Societal Impact:**

lack of understanding for each of the three terms

**Main Review:**

All the techniques seem well motivated, but their combination seems very heuristic. Results seem convincing.

My main concern is: the final loss is composed of three terms, but there is not much ablation study to understand the importance of each term. There is some ablation study in terms of adding mixup or not, but we don’t know how the later two terms have helped.

Clarity needs improvement.
* notations are a bit confusing. For example, in Eq. (1), what does argmax of an indicator function mean? Did you perhaps miss Prob()?
* Function $\phi$ was used as softmax, then later was again used as the activated channel map function
* there are many tables, but no highlight in each table, hard to read
* line 308: “Table 3” should be something else?

Some questions:
* line 152: “Compress the hypothetical space $\mathcal{F}$ into a smaller space {$ g \circ h^T, g \in \mathcal{G}$} ” does it mean $\mathcal{F}=${$g \circ h^T, g \in \mathcal{G} $}? I guess not, but I think writing in this way is a bit confusing.
* line 274: “mixup coefficient is $\lambda=1$”? doesn’t that mean there is no mixup?
* In Table 2 caption, how is the clean accuracy “significantly higher”? I see in the left column Free-AT achieves the highest clean accuracy.
* line286: “random augmentation” was never introduced earlier, and suddenly it’s added to the proposed method and shows better results. Is random augmentation also added to the baselines?


**Time Spent Reviewing:**

6

---

> ### Author Response · Authors · 2021-08-10
> **Reply to R1**
>
> We are thankful for your time and insightful comments. We are particularly grateful for your recommendations to improve the manuscript. We answer your questions here.
>
> _____
>
> **Q1:** My main concern is: the final loss is composed of three terms, but there is not much ablation study to understand the importance of each term. There is some ablation study in terms of adding mixup or not, but we don’t know how the later two terms have helped. **Limitations**: lack of understanding for each of the three terms.
>
> **A1:** We have compared the effect of individual components in Table 6, Section A.4 of supplementary material. In summary, ACM loss alone can transfer significant robustness from the teacher; the addition of soft labels and mixup further improves this transfer. Specifically, robustness with only ACM loss is 48.38%, the addition of soft-labels improves it to 49.53%, the addition of mixup improves it to 52.29%, and the addition of both of these components make final robustness to 56.65%. For more details about these experiments, please refer to Section A.4 in the supplementary material.
>
> We also discussed the role of low, mid, and high-level features in ACM loss in the response of R4. We recapitulate the discussion here: mid-level features seem more crucial for robustness and high-level features for clean accuracy. However, using all levels of features improve overall performance.
>
> ____
>
> **Q2:**  line 152: “Compress the hypothetical space F into a smaller space ${g∘h^T,g∈G}$​​ ” does it mean $F={g∘h^T, g∈G}$​​? I guess not, but I think writing in this way is a bit confusing.
>
> **A2:** The notation $\mathcal F$​​ stands for a general hypothetical space, i.e., $\mathcal F = \{ g \circ h^T, \,\, g \in \mathcal G, h \in \mathcal H\}$​​. Here $\mathcal H$​​ is the set of all feature extractors. When a teacher feature extractor is given, i.e., $h$​​ is fixed, we can obtain a smaller hypothetical space $\{ g \circ h^T, g \in \mathcal G \}$​​.
>
> ____
>
> **Q3:**  “mixup coefficient is $\lambda=1$​”? doesn’t that mean there is no mixup?
>
> **A3:**  Thank you for pointing this out. It should have been $\alpha$​​​​ or mixup hyperparameter as noted in Section A.1 of supplementary material. Following the mixup paper, we have used $\alpha \in (0, \infty)$​​​​ and $\lambda \sim Beta(\alpha, \alpha)$​​​​.
>
> ____
>
> **Q4:**  In Table 2 caption, how is the clean accuracy “significantly higher”?
>
> **A4:** Our method achieves better clean accuracy for higher $\epsilon$​​​​, e.g., for $\epsilon=4$​​​​, 62.05% compared to 55.45% for Fast-AT and 60.21% for Free-AT. However, we agree that the statement in the caption is slightly misleading since our method does not achieve better clean accuracy for $\epsilon=2$​​​​. To accurately reflect the results, we have modified the caption as follows: “Our method has significantly better robustness while also achieving comparable or better clean accuracy."
>
> ___
>
> **Q5:** Line286: “random augmentation” was never introduced earlier, and suddenly it’s added to the proposed method and shows better results. Is random augmentation also added to the baselines?
>
> **A5:**  We used random augmentation only for the ImageNet experiments in Table 2. Please note that the robustness of our method is already comparable to adversarial training methods even without using random augmentation. For instance, for test $\epsilon=2$​​​​, MixACM has a robustness of 45.96% compared to 43.46% and 43.39% for Fast-AT and Free-AT, respectively. The addition of random augmentation further improves our results. For comparisons, we have used results given in the Free-AT and Fast-AT papers.
>
> ___
>
> **Q6:** No highlight in each table, hard to read.
>
> **A6:** It is difficult to highlight better results in the tables as there is no single uniform criterion to quantify superior performance. For example, Table 1 compares our method with existing robustness transfer methods and adversarial training methods against three metrics: accuracy, robustness, and whether a method uses back-propagation. The ACM is better than robustness transfer methods for all of these metrics. Compare to state-of-the-art adversarial training methods, it is more efficient, has better clean accuracy, and has competitive robustness. However, highlighting all of this is hard. Therefore, we put the explanation in the captions rather than highlighting them via numbers.
>
> ___
>
> **Q7:** Function $\phi$ was used as softmax, then later was again used as the activated channel map function?
>
> **A7:** To distinguish between the two functions, we used different subscripts (softmax: $\phi_{\gamma}$​ and ACM: $\phi_c$​). However, to improve the clarity, We have replaced $\phi_c$​ with $g_c(.)$​ in the manuscript.
> ___
>
> **Q8:** Notations are a bit confusing. For example, in Eq. (1), what does argmax of an indicator function mean? Did you perhaps miss Prob()?
>
> **A8:** Sorry for the typo. The 'argmax' should be 'max'. In Eq. (1), the indicator is one when $\hat y$​ is incorrect. The expectation of the indicator function represents the probability of misclassification. The operator 'max' checks the neighborhood of $x$​.  Therefore, the expectation of the maximum of the indicator represents the adversarial accuracy.
> ___
>
> **Q9:** line 308: “Table 3” should be something else?
>
> **A9:**  Thank you for pointing this typo out. It should have been Figure 3. We have modified the manuscript to rectify this mistake.
>
> ____

---

> > ### Comment · Reviewer_Fo9x · 2021-08-30
> > **increase score to 6**
> >
> > Thanks for the authors' detailed response. The content in supplementary addressed my main concern. So I would like to increase my score to 6. I think it would be a good idea to mention  in the main text what's in the supplementary and maybe also briefly summarize the main points. I would also encourages the authors to thoroughly proof-read the paper to improve clarity and fix all typos.

---

> > > ### Author Response · Authors · 2021-08-30
> > > **Thank You for Increasing the Score**
> > >
> > > Thank you very much for increasing the score. We will summarize the main findings of ablation studies in the Experiments section. We are also modifying the manuscript to fix typos and other minor issues.

---

### Author Response · Authors · 2021-08-10
**General Response**

We thank reviewers for their time and valuable and positive feedback.

We are encouraged that the reviewers found our method well-motivated (**R1**), novel (**R2**), with very interesting motivation, and having theoretical analysis that well explains the principles behind it (**R3**). We are happy that they found our paper well-written and easy to follow (**R2**), having relatively good writing and easy to understand (**R3**), and well-written (**R4**). We are also pleased that they regarded our experiments as having convincing results (**R1**), extensive and showing improvements (**R2**), demonstrating outstanding performance (**R3**), solid and extensive, and setting a new benchmark in robustness distillation (**R4**).

One main concern was the lack of ablation studies. Our paper has multiple ablation studies in the supplementary material (Section A). We also have conducted a few more experiments to answer the questions raised here. We are happy to answer any further queries.

**Reviewer Acronym Reference:** We have used the order of the reviews as the reviewer's identity. Specifically, we have used the following: Fo9x as R1, d7Ao as R2, 8Aug as R3, and up6G as R4.

---

### Decision · Program_Chairs · 2021-09-27

**Decision:**

Accept (Poster)

**Comment:**

This paper describes a method (MixACM) for transferring robustness from a teacher model to a student mode using activated channel map matching and mix-up, obviating the need for expensive adversarial training.

A key concern raised by reviewers was that, while the individual components of the method are well motivated and the overall combination of them appears to be highly effective, these individual components are already known, and their simple combination is not as novel or theoretically motivated as one might hope. Reviewers also expressed concerns about quality of writing and the overall clarity of the paper.

This is something of a borderline case, but given the strong results produced by this method, I nonetheless recommend that the work be accepted.